# TimeSqueeze: Dynamic Patching for Efficient Time Series Forecasting

## Abstract

Recent progress in time series forecasting has produced large foundation models with strong generalization across domains. However, many of these models rely on transformer backbones, making their effectiveness constrained by the cost of processing the input context. The quadratic computational complexity with respect to sequence length imposes a fundamental trade-off on existing designs: they must either preserve high-frequency information using point-wise embeddings, which is computationally expensive for long sequences, or employ patch-based embeddings to reduce sequence length at the risk of discarding critical temporal details. To overcome this limitation, we present TimeSqueeze, a hybrid forecasting architecture that combines the strengths of both point and patch embeddings through **dynamic time series compression**. TimeSqueeze introduces a novel two-stage hybrid representation: (1) a *lightweight state-space encoder* processes the full-resolution time series with point-wise embeddings to extract fine-grained temporal features, and (2) an adaptive patching module intelligently prunes these features using *variable-sized patches*, assigning smaller patches to information-rich regions and larger patches to redundant segments. This hybrid approach yields a variable-resolution representation that preserves critical temporal details while reducing computational overhead. By retaining the fidelity of point embeddings and the efficiency of patch embeddings, the resulting compressed sequence enables the Transformer backbone to substantially reduce the input length without sacrificing forecasting accuracy. Extensive experiments demonstrate that TimeSqueeze achieves state-of-the-art forecasting performance while delivering substantial computational advantages, including up to $8\times$ improvement in pretraining data efficiency and up to $20\times$ reduction in pretraining time compared to equivalent point-embedding models.

## 1 Introduction

Accurate time-series forecasting is crucial across numerous domains, including energy, finance, climate, and healthcare. Historically, forecasting has relied on narrow, task-specific statistical models; however, recent advances in deep learning have enabled the development of versatile, generalist models capable of cross-domain transfer. In particular, time-series foundation models trained on heterogeneous datasets offer flexible zero-shot and few-shot generalization across a wide range of forecasting tasks.

Effective pretraining of these foundation models necessitates modeling long historical contexts, often extending to thousands of timesteps, which creates formidable computational and memory constraints. Recent studies demonstrate that increasing context length during pretraining yields substantial improvements in downstream inference performance (Gao et al., 2024; Liu et al., 2024). Therefore, designing architectures that remain scalable and computationally efficient under long-context regimes is imperative for realizing the full potential of time series foundation models.

Central to addressing these scalability challenges is the design of an efficient tokenizer that effectively represents input signals in an embedding space while managing computational complexity. Current approaches predominantly adopt one of two strategies. The first approach involves independently encoding each time point (Zhou et al., 2021; Wu et al., 2021; Zhou et al., 2022; Ansari et al., 2024; Shi et al., 2024), which preserves fine-grained temporal variations and accommodates data of arbitrary frequency and seasonality. However, this point-wise encoding strategy suffers from limited scalability as sequence length increases, which is precisely the bottleneck that impedes long-context pretraining. The second approach, pioneered by Nie et al. (2022) and subsequently adopted by numerous transformer-based forecasting models (Goswami et al., 2024; Das et al., 2024; Woo et al.,

2024; Liu et al., 2024), employs fixed-size patching to compress multiple consecutive time points into a single embedding. While this patching strategy significantly enhances computational scalability, it introduces some limitations that compromise its effectiveness. First, determining the optimal patch size is non-trivial and heavily dependent on dataset-specific characteristics such as sampling frequency and seasonal patterns, typically requiring empirical evaluation across different patch sizes for each dataset. Second, and perhaps more critically, many time series exhibit heterogeneous information density across different temporal regions, with some segments displaying rapid variations while others remaining relatively stable. This temporal heterogeneity renders uniform patching suboptimal, as it fails to adapt the representational granularity to the local complexity of the signal.

Motivated by these requirements, we propose TimeSqueeze, a hybrid time-series foundation model that combines the expressive power of point-embeddings with the computational efficiency of patch-embeddings. First, a lightweight state-space encoder extracts local fine-grained features at full resolution. Then, a dynamic patching module groups these embeddings into patches of varying sizes, allocating smaller patches to information-rich regions and larger patches to redundant ones, yielding a variable-resolution representation. This compressed sequence is processed by a Transformer backbone, which operates on significantly fewer tokens while preserving salient temporal dynamics, thereby overcoming fixed patch size limitations and enabling scalable, high-fidelity modeling.

Our contributions are as follows:

- We propose TimeSqueeze, the first hybrid forecasting architecture to incorporate dynamic, content-aware patching for adaptive compression in time series.
- We demonstrate that TimeSqueeze integrates seamlessly with existing Transformer backbones (e.g., Time-MoE), enabling pretraining of large-scale time series foundation models with substantially reduced training budgets.
- We validate TimeSqueeze across diverse zero-shot forecasting benchmarks, achieving performance on par with state-of-the-art point embedding models while delivering up to $20\times$ faster training and $10\times$ faster inference.

## 2 RELATED WORKS.

**Long-sequence architectures.** While Transformer architectures (Vaswani et al., 2017) have shown strong time series forecasting performance due to their expressivity and flexibility, their quadratic computational and memory complexity with respect to sequence length limits their scalability to long historical contexts. Innovations such as (Li et al., 2019; Wu et al., 2021; Zhou et al., 2021) have adapted Transformers for long-term forecasting, but pretraining on extremely long contexts remains challenging. Recently, time-series foundation models have demonstrated scalability to long contexts supporting arbitrary forecasting horizons, while Time-MoE (Shi et al., 2024) leveraged Mixture-of-Experts routing to enable the first billion-parameter model with tractable inference. Despite these advances, the cost of long-context pretraining remains high due to the underlying Transformer backbone. Although state space model (SSM) architectures Gu & Dao (2023) handle long contexts more efficiently, they remain underexplored for time series forecasting, highlighting the need for scalable methods for efficient long-context processing.

**Patch-based compression.** Introduced in PatchTST (Nie et al., 2022), patch-based compression has emerged as a fundamental technique for scaling time series foundation models. By embedding contiguous sub-sequences (patches) rather than individual time points, this approach reduces the effective sequence length while preserving essential local temporal patterns. Subsequent foundation models, including TimesFM (Das et al., 2024), Moment (Goswami et al., 2024), Moirai (Woo et al., 2024), and Timer-XL (Liu et al., 2024), have adopted this paradigm, collectively demonstrating that patching enables more efficient training and inference. However, these approaches utilize a fixed patch size for a given sequence, limiting their application to real-world data with high temporal variance, underscoring the need for dynamic, data-driven compression strategies that can adjust patching to varying temporal structures within a series.

**Insights from language modeling.** Similar challenges arise in large language models (LLMs), where the choice of input representation has a direct impact on scalability and fidelity. Conventional tokenization introduces systematic biases and brittle dependencies, motivating tokenizer-free models that operate at the byte level. Yet, naïve byte-level processing leads to prohibitively long input sequences

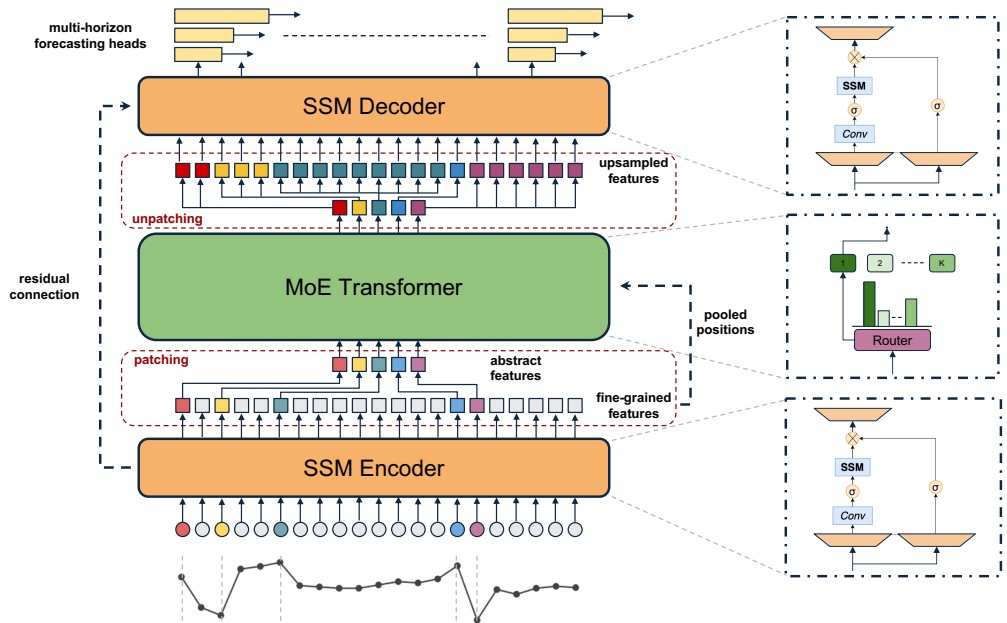

Figure 1: Architectural overview of TimeSqueeze. An SSM encoder first processes the raw series at full resolution to extract fine-grained features. Dynamic patching then adaptively compresses the sequence, selecting the salient subset of embeddings. A Transformer backbone performs contextual modeling on the downsampled features, and an unpatching module upsamples the signal to the original resolution while preserving causality. Finally, an SSM decoder combines the compressed and fine-grained features, passing the hybrid features to multi-horizon heads, thereby improving efficiency without sacrificing temporal fidelity.

(Slagle, 2024), straining attention-based architectures. To overcome this, adaptive compression techniques have been proposed. The Byte Latent Transformer (BLT) dynamically merges predictable byte spans into compact latent tokens using entropy-guided segmentation (Pagnoni et al., 2024), while H-Net (Hwang et al., 2025), inspired by U-Net (Ronneberger et al., 2015) and its broad adaptation in vision (Child, 2021; Ho et al., 2020; Wu et al., 2025), compresses and reconstructs sequences in various resolutions, and uses a state-space model for more efficient byte-level processing. These approaches highlight a key principle: efficiency and accuracy can be jointly achieved by allocating higher granularity to information-dense regions and applying more aggressive compression where redundancy dominates.

## 3 METHODOLOGY

**Problem Statement.** The fundamental objective in time-series forecasting is to predict future values based on historical observations. Given a sequence of $T$ historical data points, $\mathbf{X}_{1:T} = (x_1, x_2, \ldots, x_T) \in \mathbb{R}^T$, the goal is to estimate the next $H$ values of the series. This is formalized via a model $f_\theta$ that maps the historical context to future predictions, i.e., $\hat{\mathbf{X}}_{T+1:T+H} = f_\theta(\mathbf{X}_{1:T}) \in \mathbb{R}^H$. Adopting the channel independence principle of Nie et al. (2022), the model can flexibly process multivariate time series by decomposing inputs into collections of univariate series. This general formulation enables time-series foundation models to address forecasting tasks with arbitrary input dimensionality, thereby supporting broad applicability across diverse, real-world domains.

### 3.1 ARCHITECTURAL OVERVIEW

To combine the expressivity of point-embeddings with the computational efficiency of patch embeddings, TimeSqueeze employs a hybrid multi-resolution architecture with four key components: (1) a lightweight encoder-decoder pair operating at full input resolution to capture fine-grained local features, (2) adaptive patching modules that dynamically select salient features for efficient downsampling and upsampling, (3) a decoder-only MoE Transformer backbone for modeling causal

dependencies at scale, and (4) a multi-horizon forecasting head that jointly optimizes predictions across multiple time horizons to support both short- and long-term forecasting, as shown in Figure 1.

Formally, the end-to-end model can be described as

$$H_{1:T} = \mathcal{E}(X_{1:T}), \quad Z_{1:P} = \mathcal{M}(\mathcal{P}(H_{1:T})), \quad Y_{1:T} = \mathcal{D}(H_{1:T}, \mathcal{U}(Z_{1:P})), \quad (1)$$

where $\mathcal{E}$ is the encoder, $\mathcal{P}$ is the patching module, $\mathcal{M}$ is the MoE Transformer backbone, $\mathcal{U}$ is the unpatching module, and $\mathcal{D}$ is the decoder. Here, $X_{1:T} \in \mathbb{R}^T$ denotes the original input sequence, $H_{1:T} \in \mathbb{R}^{T \times D}$ denotes the $D$-dimensional encoder embeddings, $Z_{1:P} \in \mathbb{R}^{P \times D}$ denotes the patch-level latent representation after $\mathcal{P}$ and $\mathcal{M}$, and $Y_{1:T} \in \mathbb{R}^{T \times D}$ denotes the decoder embeddings that serve as the final representation for downstream forecasting.

### 3.1.1 STATE-SPACE ENCODER AND DECODER

The encoder and decoder modules operate directly on input time series at native resolution to preserve fine-grained temporal details essential for accurate forecasting, particularly in high-frequency data. To handle long, uncompressed sequences efficiently while generating representations suitable for subsequent patching, both modules are constructed using Mamba layers (Gu & Dao, 2023).

Mamba offers nearly linear computational scaling with respect to sequence length, enabling extraction of intricate local patterns from extended contexts by the encoder, without the quadratic complexity of traditional Transformer architectures. Further, the decoder uses the same architecture to efficiently combines outputs from the Transformer backbone with residual embeddings from the encoder to produce final representations for forecasting, creating a rich multi-scale feature space that captures both local fine-grained patterns and global contextual dependencies.

### 3.1.2 DYNAMIC PATCHING AND UNPATCHING

After the encoder produces fine-grained representations, the patching module compresses the sequence of embeddings before passing them to the Transformer backbone. The objective is to allocate computational resources efficiently by employing a dynamic patching strategy that adapts to the local complexity of the input signal. This strategy forms larger patches to compress regions of low information density while using smaller patches to preserve detail in regions of high information content. A visualization of the patch boundaries for different datasets is provided in Appendix H.

**Patching.** Unlike language models that operate on discrete token sequences, time series data exist in continuous space and exhibit rich statistical properties. This continuous nature makes time series particularly amenable to characterization via statistical measures such as local variance or power, without relying on external metrics for guidance (Pagnoni et al., 2024). We leverage this by tracking the absolute difference between consecutive samples, comparing it to the average signal power within a predetermined lookback window, and then computing the patch boundaries in the original signal space rather than the embedding space. Formally, we maintain a sliding window $\mathcal{W}_i = \{x_{i-L}, \ldots, x_{i-1}\}$ of length $L$ to compute the local average power as

$$P_i = \frac{1}{L} \sum_{j=i-L}^{i-1} x_j^2.$$

Our adaptive patching mechanism declares a patch boundary at timestep $i$ if the absolute difference between consecutive samples exceeds a threshold scaled by the local power, which we refer to as **relative deviation-based patching**:

$$b_i = \begin{cases} 1 & \text{if } |x_i - x_{i-1}| > \tau\sqrt{P_i} \\ 0 & \text{otherwise} \end{cases}.$$

Here, $\tau > 0$ is a tunable threshold parameter controlling patch sensitivity and the average compression ratio. Using $\sqrt{P_i}$ normalizes the threshold with respect to signal amplitude, allowing the method to adapt dynamically across varying signal magnitudes and variances. Once patch boundaries are determined, the embeddings within each patch are compressed by retaining only the boundary embeddings and discarding intermediate ones (Figure 1). Note that retaining only the boundary embeddings helps preserve causality for the subsequent unpatching step.

**Unpatching.** The unpatching module restores the compressed embeddings to the original sequence length while maintaining causal consistency. After backbone processing of boundary embeddings, each updated embedding is repeated across all timesteps within its corresponding patch. Since boundary embeddings represent the start of each patch, the reconstructed output at timestep $t$ depends only on inputs from times $\leq t$, preventing leakage of future information.

**Positional Information.** Unlike language models, which predict the next discrete token, time series forecasting models demands more nuanced objective during pretraining. Forecasting must occur at a specified frequency within the original continuous signal space, not within the compressed embedding space. Prior works on tokenizer-free language modeling, such as BLT (Pagnoni et al., 2024) and Dynamic Chunking (Hwang et al., 2025), do not retain the original positional indices and restrict the attention mechanism to relative positional information post-downsampling. In contrast, TimeSqueeze explicitly preserves the position IDs of embeddings before downsampling and utilizes these absolute positions to compute attention after compression.

### 3.1.3 MIXTURE-OF-EXPERTS TRANSFORMER BACKBONE

Due to its modular design, our hybrid feature extraction framework is compatible with any existing time-series forecasting backbone. In this work, we adopt the Time-MoE backbone (Shi et al., 2024), a scalable decoder-only Transformer augmented with a sparse MoE routing mechanism. Time-MoE incorporates several enhancements to improve training stability and forecast accuracy: it employs RMSNorm for layer normalization and replaces absolute positional encodings with Rotary Positional Embeddings (RoPE), facilitating better handling of variable sequence lengths and improved extrapolation. Following established design patterns, the standard feed-forward network (FFN) is replaced by an MoE layer containing a pool of $N$ non-shared experts alongside one shared expert that consolidates common knowledge. For each input token, a routing mechanism selects the top $K$ non-shared experts to process the signal, enabling efficient scaling to billions of parameters while maintaining manageable inference costs.

### 3.1.4 MULTI-HORIZON FORECASTING

To enhance forecasting flexibility and robustness, we employ a multi-horizon forecasting head as introduced in (Shi et al., 2024). This approach enables simultaneous prediction across multiple future horizons rather than restricting the model to a single forecast length. Specifically, it consists of multiple single-layer FFNs, each dedicated to a distinct forecasting horizon. The model is trained using a composite loss aggregating errors from all horizons, which improves generalization. During inference, a simple scheduling strategy selects the appropriate horizon-specific output, enabling the model to produce forecasts of arbitrary length flexibly.

## 3.2 MODEL TRAINING

**Pretraining Dataset.** Efficient pretraining of a foundation model necessitates a large and diverse dataset. For this purpose, we employ the Time-300B dataset (Shi et al., 2024), a high-quality, open-access dataset composed of time series from numerous public sources across various sectors, including weather, transportation, and finance, which is further expanded with synthetic data. It consists of a broad range of frequencies, ranging from seconds to yearly, and a massive scale of over 300 billion time points, making it well-suited for pretraining large-scale models.

**Loss Formulation.** Following Shi et al. (2024), our training objective is a composite loss function that combines a primary forecasting loss with an auxiliary term for load balancing, which enables a fair comparison against the point-embedding baseline Time-MoE. The primary auto-regressive loss, $\mathcal{L}_{\text{ar}}$, is the Huber Loss (Huber, 1992), chosen for its robustness against outliers:

$$\mathcal{L}_{ar}(x_t, \hat{x}_t) = \begin{cases} \frac{1}{2}(x_t - \hat{x}_t)^2, & \text{if } |x_t - \hat{x}_t| \leq \delta, \\ \delta\left(|x_t - \hat{x}_t| - \frac{1}{2}\delta\right), & \text{otherwise,} \end{cases} \quad (2)$$

where $\delta$ is a hyperparameter that balances the quadratic ($L_2$) and linear ($L_1$) penalties.

To ensure balanced expert utilization and prevent routing collapse, we incorporate an auxiliary loss, $\mathcal{L}_{\mathrm{aux}}$, as proposed in Fedus et al. (2022):

$$\mathcal{L}_{\mathrm{aux}} = N \sum_{i=1}^{N} f_i r_i, \tag{3}$$

where $f_i$ is the fraction of tokens dispatched to expert $i$, and $r_i$ is the average router probability assigned to the expert. The final training loss, $\mathcal{L}$, averages the auto-regressive loss across $K$ multi-resolution projections and combines it with the weighted auxiliary loss:

$$\mathcal{L} = \frac{1}{K} \sum_{j=1}^{K} \mathcal{L}_{\mathrm{ar}} \left( \mathbf{X}_{t+1:t+p_j}, \hat{\mathbf{X}}_{t+1:t+p_j} \right) + \alpha \mathcal{L}_{\mathrm{aux}}, \tag{4}$$

where $p_j$ is the forecast horizon for the $j$-th projection and $\alpha$ is a scaling coefficient.

**Model Configuration.** We consider two model sizes in this work, demonstrating the scalability of our approach. TimeSqueeze $_{\mathrm{base}}$ has a total of 117M parameters with 54M active parameters, while TimeSqueeze $_{\mathrm{large}}$ contains 469M total parameters with 216M active parameters. Both models are trained for 100,000 steps with a batch size of 256 and a maximum context length of 2048, corresponding to 500K time points per iteration and a total of 50B time steps during pretraining. Finally, for the patching and unpatching modules, we target an average compression rate of 4 in TimeSqueeze by setting the threshold factor $\tau = 0.3$, and limiting the maximum patch size to 8, balancing computational savings and information preservation. Further configuration details are provided in Appendix A.

## 4 EXPERIMENTAL RESULTS

**Baselines.** Our primary objective is to demonstrate the efficiency and performance improvements of TimeSqueeze over point embedding models through dynamic context compression. We use TimeMoE as our baseline, and pretrain TimeSqueeze following the training scheme of Shi et al. (2024), but using $8\times$ lesser data and $\approx 20\times$ less train time, as shown in Figure 2a. We forecast on four prediction horizons $\{96, 192, 336, 720\}$ but use the same context length of 512 in all cases. While, we study the point-forecasting performance of TimeSqueeze, but it can easily be extended to provide probabilistic forecasts by substituting the model's linear projection head with a probabilistic head. We assess model performance using the mean squared error (MSE) and mean absolute error (MAE), computed between the predicted values and the ground truth. For completeness, we also compare against Moirai-large (Woo et al., 2024), TimesFM (Das et al., 2024), Moment (Ansari et al., 2024), and Chronos (Goswami et al., 2024), with results taken from Shi et al. (2024).

### 4.1 ZERO-SHOT FORECASTING

We first compare the zero-shot performance of TimeSqueeze $_{\mathrm{base}}$ and TimeSqueeze $_{\mathrm{large}}$ against Time-MoE$_{\mathrm{base}}$ and $_{\mathrm{large}}$ on the well-studied long-term forecasting benchmarks (Zhou et al., 2021) and the Weather data (Wu et al., 2021). These datasets were not included in the Time-300B dataset and not used for training the TimeSqueeze. Detailed zero-shot forecasting results are presented in Table 1, demonstrating that TimeSqueeze performs remarkably well, achieving a performance similar to that of Time-MoE. Further results for higher compression rates are provided in Appendix D.

Additional comparisons for TimeSqueeze against Time-MoE are presented in Section E. We note that the performance of TimeSqueeze $_{\mathrm{large}}$ is slightly worse than TimeSqueeze $_{\mathrm{base}}$ in some scenarios, likely due to the limited training budget.

### 4.2 IN-DISTRIBUTION FORECASTING

We now measure the full-shot performance by finetuning TimeSqueeze on the train split of the same benchmarks. For finetuning, we choose a learning rate of 1e-4 and fine-tune the pretrained model for just one epoch. We compare the full-shot performance against Liu et al. (2023); Wang et al. (2024); Wu et al. (2022); Nie et al. (2022); Zeng et al. (2023), in addition to the finetuned version of Time-MoE$_{\mathrm{base}}$. As seen from Table 2, TimeSqueeze still performs close to Time-MoE, and outperforms all other baselines considered.

Table 1: Performance comparison of zero-shot forecasting. **Bold** for best and underscore for 2nd best.

| Models | Metrics | TimeSqueeze _base_ | | TimeSqueeze _large_ | | Time-MoE_base_ | | Time-MoE_large_ | | Moirai_base_ | | TimesFM | | Moment | | Chronos_large_ | |
|---|---|---|---|---|---|---|---|---|---|---|---|---|---|---|---|---|---|
| | | MSE | MAE | MSE | MAE | MSE | MAE | MSE | MAE | MSE | MAE | MSE | MAE | MSE | MAE | MSE | MAE |
| ETTh1 | 96 | 0.359 | 0.385 | 0.360 | **0.379** | 0.357 | 0.381 | **0.350** | 0.382 | 0.376 | 0.392 | 0.414 | 0.404 | 0.688 | 0.557 | 0.441 | 0.390 |
| | 192 | 0.400 | 0.410 | 0.402 | 0.407 | 0.388 | 0.412 | **0.384** | 0.404 | 0.417 | 0.413 | 0.465 | 0.434 | 0.688 | 0.560 | 0.502 | 0.424 |
| | 336 | 0.420 | 0.423 | 0.423 | **0.412** | **0.411** | 0.430 | **0.411** | 0.434 | 0.433 | **0.428** | 0.503 | 0.456 | 0.675 | 0.563 | 0.576 | 0.467 |
| | 720 | 0.428 | 0.446 | 0.441 | 0.448 | **0.427** | 0.455 | 0.449 | 0.477 | 0.447 | **0.444** | 0.511 | 0.481 | 0.683 | 0.585 | 0.835 | 0.583 |
| | Avg. | 0.402 | 0.416 | 0.407 | 0.414 | **0.394** | 0.419 | 0.400 | 0.420 | 0.417 | **0.419** | 0.473 | 0.443 | 0.683 | 0.566 | 0.588 | 0.466 |
| ETTh2 | 96 | **0.282** | 0.346 | 0.290 | 0.355 | 0.305 | 0.359 | 0.302 | 0.354 | 0.294 | **0.330** | 0.315 | 0.349 | 0.342 | 0.396 | 0.320 | 0.345 |
| | 192 | **0.349** | 0.394 | 0.368 | 0.413 | 0.351 | 0.386 | 0.364 | 0.385 | 0.365 | **0.375** | 0.388 | 0.395 | 0.354 | 0.402 | 0.406 | 0.399 |
| | 336 | 0.379 | 0.422 | 0.405 | 0.447 | 0.391 | 0.418 | 0.417 | 0.425 | 0.376 | **0.390** | 0.422 | 0.427 | **0.356** | 0.407 | 0.492 | 0.453 |
| | 720 | 0.444 | 0.471 | 0.445 | 0.441 | 0.419 | 0.454 | 0.537 | 0.496 | 0.416 | **0.433** | 0.443 | 0.454 | **0.395** | 0.434 | 0.603 | 0.511 |
| | Avg. | 0.363 | 0.408 | 0.377 | 0.414 | 0.366 | 0.404 | 0.405 | 0.415 | 0.362 | **0.382** | 0.392 | 0.406 | **0.361** | 0.409 | 0.455 | 0.427 |
| ETTm1 | 96 | 0.312 | 0.344 | **0.304** | **0.334** | 0.338 | 0.368 | 0.309 | 0.357 | 0.363 | 0.356 | 0.361 | 0.370 | 0.654 | 0.527 | 0.457 | 0.403 |
| | 192 | 0.372 | 0.385 | 0.358 | **0.367** | 0.353 | 0.388 | **0.346** | 0.381 | 0.388 | 0.375 | 0.414 | 0.405 | 0.662 | 0.532 | 0.530 | 0.450 |
| | 336 | 0.435 | 0.425 | 0.403 | **0.396** | 0.381 | 0.413 | **0.373** | 0.408 | 0.416 | **0.392** | 0.445 | 0.429 | 0.672 | 0.537 | 0.577 | 0.481 |
| | 720 | 0.547 | 0.494 | 0.486 | 0.444 | 0.504 | 0.493 | 0.475 | 0.477 | 0.460 | **0.418** | 0.512 | 0.471 | 0.692 | 0.551 | 0.660 | 0.526 |
| | Avg. | 0.417 | 0.412 | 0.388 | **0.385** | 0.394 | 0.415 | **0.376** | 0.406 | 0.406 | **0.385** | 0.433 | 0.418 | 0.670 | 0.536 | 0.555 | 0.465 |
| ETTm2 | 96 | 0.181 | 0.275 | **0.179** | 0.272 | 0.201 | 0.291 | 0.197 | 0.286 | 0.205 | 0.273 | 0.202 | **0.270** | 0.260 | 0.335 | 0.197 | 0.271 |
| | 192 | **0.248** | 0.323 | 0.251 | 0.325 | 0.258 | 0.334 | 0.250 | 0.322 | 0.275 | 0.316 | 0.289 | 0.321 | 0.289 | 0.350 | 0.254 | **0.314** |
| | 336 | **0.310** | 0.363 | 0.319 | 0.368 | 0.324 | 0.373 | 0.337 | 0.375 | 0.329 | 0.350 | 0.360 | 0.366 | 0.324 | 0.369 | 0.313 | **0.353** |
| | 720 | 0.431 | 0.437 | 0.425 | 0.428 | 0.488 | 0.464 | 0.480 | 0.461 | 0.437 | 0.411 | 0.462 | 0.430 | **0.394** | **0.409** | 0.416 | 0.415 |
| | Avg. | 0.292 | 0.349 | 0.294 | 0.348 | 0.317 | 0.365 | 0.316 | 0.361 | 0.311 | **0.337** | 0.328 | 0.346 | 0.316 | 0.365 | 0.295 | **0.338** |
| Weather | 96 | 0.1671 | 0.2172 | 0.170 | 0.221 | **0.160** | 0.214 | **0.159** | 0.213 | 0.220 | 0.217 | - | - | 0.243 | 0.255 | 0.194 | 0.235 |
| | 192 | 0.2188 | 0.2690 | 0.224 | 0.275 | **0.210** | 0.260 | 0.215 | 0.266 | 0.271 | 0.259 | - | - | 0.278 | 0.329 | 0.249 | 0.285 |
| | 336 | 0.278 | 0.315 | 0.292 | 0.326 | **0.274** | 0.309 | 0.291 | 0.322 | 0.286 | 0.297 | - | - | 0.306 | 0.346 | 0.302 | 0.327 |
| | 720 | 0.364 | 0.372 | 0.409 | 0.396 | 0.418 | 0.405 | 0.419 | 0.400 | 0.373 | 0.354 | - | - | **0.350** | 0.374 | 0.372 | 0.378 |
| | Avg. | **0.257** | 0.293 | 0.274 | 0.305 | 0.265 | 0.297 | 0.271 | 0.300 | 0.287 | 0.281 | - | - | 0.294 | 0.326 | 0.279 | 0.306 |
| **Average** | | **0.346** | 0.376 | 0.348 | 0.373 | 0.347 | 0.380 | 0.352 | 0.380 | 0.357 | **0.361** | 0.407 | 0.403 | 0.465 | 0.440 | 0.434 | 0.400 |

## 4.3 Efficiency Comparison

We now compare the training and inference efficiency of TimeSqueeze _base_ with the point-embedding baseline Time-MoE_base_ model in terms of GPU hours and memory utilization. All experiments were conducted on $2\times$ NVIDIA A100 80GB GPUs.

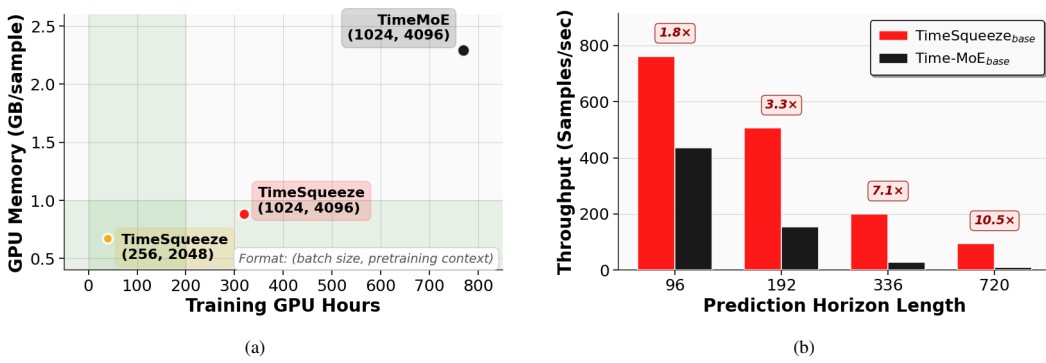

| (a) | (b) |

Figure 2: Computational efficiency comparison between TimeSqueeze _base_ and Time-MoE: (a) Training memory and time requirements across different batch sizes and context lengths. TimeSqueeze achieves comparable performance while reducing memory usage by $3.4\times$ and training time by $\approx 20\times$. (b) Inference throughput across prediction horizons. TimeSqueeze delivers up to $10.5\times$ higher throughput for longer prediction horizons.

In Figure 2a, we plot the pretraining time and memory required for different (batch size, context length) for Time-MoE and TimeSqueeze, when trained for $100,000$ iterations. When using $(1024, 4096)$, we see that TimeSqueeze uses $2.6\times$ less memory and $2.4\times$ less compute compared to Time-MoE. Furthermore, when running on a smaller budget, TimeSqueeze is trained with $(256, 2048)$, which uses $3.4\times$ less memory and $19.25\times$ less training time while still achieving performance comparable to Time-MoE, as shown in Table 1.

In Figure 2b, we plot the inference throughput for different forecasting horizons. We use a context length of $512$ for TimeSqueeze and the original context lengths from (Shi et al., 2024) for Time-MoE. We see that TimeSqueeze scales more gracefully with respect to context length, showing up to $10.5\times$

Table 2: Performance comparison of full-shot forecasting. **Bold** for best and underscore for 2nd best.

| Models | Metrics | TimeSqueeze_base | | Time-MoE_base | | iTransformer | | TimeMixer | | TimesNet | | PatchTST | | DLinear | |
|---|---|---|---|---|---|---|---|---|---|---|---|---|---|---|---|
| | | MSE | MAE | MSE | MAE | MSE | MAE | MSE | MAE | MSE | MAE | MSE | MAE | MSE | MAE |
| ETTh1 | 96 | 0.354 | 0.384 | **0.345** | **0.375** | 0.386 | 0.405 | 0.375 | 0.400 | 0.384 | 0.402 | 0.414 | 0.419 | 0.423 | 0.448 |
| | 192 | 0.397 | 0.412 | **0.372** | **0.396** | 0.441 | 0.436 | 0.436 | 0.429 | 0.421 | 0.429 | 0.460 | 0.445 | 0.471 | 0.474 |
| | 336 | 0.418 | 0.427 | **0.389** | **0.412** | 0.487 | 0.458 | 0.484 | 0.458 | 0.491 | 0.469 | 0.501 | 0.466 | 0.570 | 0.546 |
| | 720 | 0.423 | 0.454 | **0.410** | **0.443** | 0.503 | 0.491 | 0.498 | 0.482 | 0.521 | 0.500 | 0.500 | 0.488 | 0.653 | 0.621 |
| | Avg. | 0.398 | 0.419 | 0.379 | 0.406 | 0.454 | 0.447 | 0.448 | 0.442 | 0.454 | 0.450 | 0.468 | 0.454 | 0.529 | 0.522 |
| ETTh2 | 96 | **0.274** | **0.336** | 0.276 | 0.340 | 0.297 | 0.349 | 0.289 | 0.341 | 0.340 | 0.374 | 0.302 | 0.348 | 0.745 | 0.584 |
| | 192 | 0.337 | 0.379 | **0.331** | **0.371** | 0.380 | 0.400 | 0.372 | 0.392 | 0.402 | 0.414 | 0.388 | 0.400 | 0.877 | 0.656 |
| | 336 | **0.373** | 0.408 | 0.373 | **0.402** | 0.428 | 0.432 | 0.386 | 0.414 | 0.452 | 0.541 | 0.426 | 0.433 | 1.043 | 0.731 |
| | 720 | 0.417 | 0.449 | **0.404** | **0.431** | 0.427 | 0.445 | 0.412 | 0.434 | 0.462 | 0.657 | 0.431 | 0.446 | 1.104 | 0.763 |
| | Avg. | 0.350 | 0.393 | 0.346 | 0.386 | 0.383 | 0.406 | 0.364 | 0.395 | 0.414 | 0.496 | 0.386 | 0.406 | 0.942 | 0.683 |
| ETTm1 | 96 | 0.289 | **0.332** | **0.286** | 0.334 | 0.334 | 0.368 | 0.320 | 0.357 | 0.338 | 0.375 | 0.329 | 0.367 | 0.404 | 0.426 |
| | 192 | 0.344 | 0.366 | **0.307** | **0.358** | 0.377 | 0.391 | 0.360 | 0.381 | 0.374 | 0.387 | 0.367 | 0.385 | 0.450 | 0.451 |
| | 336 | 0.398 | 0.396 | **0.354** | **0.390** | 0.426 | 0.420 | 0.390 | 0.404 | 0.410 | 0.411 | 0.399 | 0.410 | 0.532 | 0.515 |
| | 720 | 0.502 | 0.451 | **0.433** | 0.445 | 0.491 | 0.459 | 0.454 | 0.441 | 0.478 | 0.450 | 0.454 | **0.439** | 0.666 | 0.589 |
| | Avg. | 0.383 | 0.386 | 0.345 | 0.381 | 0.407 | 0.409 | 0.381 | 0.395 | 0.400 | 0.405 | 0.387 | 0.400 | 0.513 | 0.495 |
| ETTm2 | 96 | **0.168** | **0.256** | 0.172 | 0.265 | 0.180 | 0.264 | 0.175 | 0.258 | 0.187 | 0.267 | 0.175 | 0.259 | 0.287 | 0.366 |
| | 192 | **0.225** | **0.298** | 0.228 | 0.306 | 0.250 | 0.309 | 0.237 | 0.299 | 0.249 | 0.309 | 0.241 | 0.302 | 0.414 | 0.392 |
| | 336 | **0.278** | **0.335** | 0.281 | 0.345 | 0.311 | 0.348 | 0.298 | 0.340 | 0.321 | 0.351 | 0.305 | 0.343 | 0.597 | 0.542 |
| | 720 | **0.366** | **0.395** | 0.403 | 0.424 | 0.412 | 0.407 | 0.391 | 0.396 | 0.408 | 0.403 | 0.402 | 0.400 | 1.730 | 1.042 |
| | Avg. | **0.259** | 0.321 | 0.271 | 0.335 | 0.288 | 0.332 | 0.275 | 0.323 | 0.291 | 0.332 | 0.280 | 0.326 | 0.757 | 0.610 |
| Weather | 96 | 0.152 | **0.199** | 0.151 | 0.203 | 0.154 | 0.208 | 0.163 | 0.209 | 0.172 | 0.220 | 0.177 | 0.218 | 0.158 | 0.230 |
| | 192 | 0.201 | 0.249 | **0.195** | **0.246** | 0.202 | 0.251 | 0.208 | 0.250 | 0.219 | 0.261 | 0.225 | 0.259 | 0.206 | 0.277 |
| | 336 | 0.259 | 0.297 | **0.247** | 0.288 | 0.252 | **0.287** | 0.251 | **0.287** | 0.280 | 0.306 | 0.278 | 0.297 | 0.272 | 0.335 |
| | 720 | 0.360 | 0.372 | 0.352 | 0.366 | **0.302** | 0.376 | 0.339 | **0.341** | 0.365 | 0.359 | 0.354 | 0.348 | 0.308 | 0.418 |
| | Avg. | 0.243 | 0.279 | **0.236** | 0.275 | 0.250 | 0.280 | 0.240 | 0.271 | 0.259 | 0.286 | 0.258 | 0.280 | 0.258 | 0.315 |
| **Average** | | 0.327 | 0.360 | **0.315** | **0.357** | 0.356 | 0.375 | 0.342 | 0.365 | 0.364 | 0.394 | 0.356 | 0.373 | 0.600 | 0.525 |

faster inference for longer prediction horizons, making TimeSqueeze more suitable for on-device inference.

## 4.4 ABLATION STUDIES

We conduct systematic ablation studies to quantify the contributions of key components in TimeSqueeze. We use TimeSqueeze _base for all ablation studies, which were trained using the same approach as described in Section 3.2. During inference, we use a context length of 512 for TimeSqueeze and the original context lengths used in (Shi et al., 2024) for Time-MoE.

### 4.4.1 MODEL COMPONENTS

*Dynamic vs. Fixed Patching.* We compare our proposed relative deviation-based dynamic patching approach with fixed patching. For the fixed patching baseline with patch size 4, embeddings are uniformly downsampled by retaining every 4th element. Results show that dynamic patching consistently outperforms fixed patching by effectively focusing computational resources on information-rich segments rather than optimizing only for a compression rate at the risk of discarding critical intermediate samples. This underscores the importance of dynamic compression strategies for handling temporal heterogeneity in time-series data.

*Mamba vs. Linear Encoder.* To assess the importance of our SSM encoder-decoder, we replace it with simple linear embedding layers akin to the architectures used in Moirai (Woo et al., 2024) and TimesFM (Das et al., 2024). The SSM-based encoder achieves substantial gains over linear projections, confirming its suitability for capturing fine-grained temporal features and its inductive bias, which is beneficial for sequential compression.

*Importance of Fine-Grained Features.* We evaluate the contribution of preserving detailed temporal information by ablating the residual connection illustrated in Figure 1, relying solely on compressed features for forecasting. This modification results in noticeable performance degradation.

*Positional Encoding Analysis.* We investigate the role of preserving positional information by comparing absolute position embeddings of boundary elements with relative positional encodings applied to compressed embeddings. Removing absolute positional cues results in notable performance drops, highlighting the necessity of absolute temporal positioning to maintain temporal coherence in the reconstructed sequences.

**Observation.** Figure 3a shows the summary of these ablations, by plotting the average MSE across the five benchmarking datasets for a prediction horizon of 96. The results clearly indicate that the inductive bias of SSM, combined with the dynamic context-aware pruning of SSM embeddings, is crucial to achieving optimal performance, while the residual connection and the use of absolute position IDs play a minor role. The full results are included in Appendix F, Table 6.

### 4.4.2 LONG-CONTEXT PRETRAINING

Recent studies show that pretraining with longer context lengths can improve inference performance even when using shorter contexts during deployment (Liu et al., 2024). We investigate this by training TimeSqueeze with different maximum pretraining context lengths under a fixed token budget of approximately 50B tokens. All models are trained for 100,000 steps, with batch sizes adjusted to account for context length differences, while maintaining an inference context length of 512 tokens.

Figure 3b demonstrates that longer pretraining contexts consistently improve inference performance even when using a shorter inference context pf 512 always. This indicates that exposure to extended sequences during pretraining enables TimeSqueeze to develop more robust temporal representations that effectively transfer to shorter inference contexts. Notably, unlike Time-MoE, TimeSqueeze achieves strong inference performance with short contexts, despite being pretrained on longer sequences, significantly reducing computational overhead during deployment.

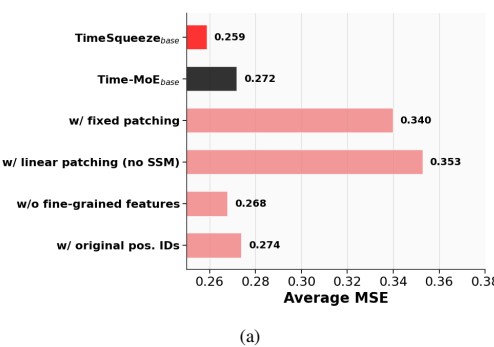
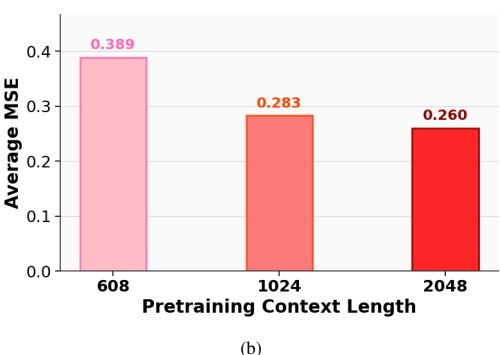

(a)                                                                    (b)

Figure 3: Model analysis: (a) Average MSE across five benchmark datasets for prediction horizon 96 with different model components. (b) Effect of Pretraining Context Length on Forecasting Performance. Longer pretraining context translates to improved performance, even when the inference context remains fixed at 512.

## 5 CONCLUSION

We present TimeSqueeze, the first time-series forecasting model to explore dynamic input compression using a content-aware patching mechanism, which combines the temporal fidelity of point embedding models with the computational efficiency of patch-based approaches. Our relative deviation-based metric enables data-driven patching, producing representations that optimally allocate computational resources to where they provide the most significant benefit for forecasting. TimeSqueeze achieves performance comparable to the baseline point embedding model Time-MoE, while achieving $8\times$ improvement in pretraining data efficiency and up to $20\times$ reduction in pretraining time.

Our work opens several promising research directions focusing on variable-rate patching and compression for time series forecasting. Despite the relative threshold-based patching being scale independent, it still requires hyperparameter tuning in terms of threshold factor $\tau$. Alternatively, patch boundaries could be learned end-to-end in embedding spaces (Hwang et al., 2025) or use an auxiliary model for guidance (Pagnoni et al., 2024). In addition, TimeSqueeze could benefit from scaling the number of parameters in the backbone and the amount of training data, similar to Time-MoE (Shi et al., 2024), while supporting multiple context compression rates with a single model.

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

# A    PRETRAINING CONFIGURATION

The training configuration follows the same as Time-MoE: forecasting horizons are set to $\{1, 8, 32, 64\}$ in the output projection, and the auxiliary loss weighting factor $\alpha$ is 0.02. We optimize with AdamW using initial learning rate $1 \times 10^{-3}$, weight decay 0.1, $\beta_1 = 0.9$, and $\beta_2 = 0.95$. The learning rate scheduler employs a linear warmup for the first 10,000 steps, followed by cosine annealing to a minimum learning rate of $5 \times 10^{-5}$. Training is performed on 2 NVIDIA A100 80GB GPUs using BF16 precision, and the configurations for each model are described in detail in Table 3.

Table 3: Model configurations.

|  | Enc. Layers | Dec. Layers | $d_{\text{model}}$ | $d_{\text{state}}$ | $d_{\text{conv}}$ | expand | Params |
|---|---|---|---|---|---|---|---|
| TimeSqueeze $_{\text{base}}$ | 2 | 2 | 384 | 128 | 4 | 4 | 4M |
| TimeSqueeze $_{\text{large}}$ | 2 | 2 | 768 | 128 | 4 | 4 | 16M |

(a) Mamba encoder–decoder.

| Model | Layers | Heads | Experts | $K$ | $d_{\text{model}}$ | $d_{\text{ff}}$ | $d_{\text{expert}}$ | Activated Params | Total Params |
|---|---|---|---|---|---|---|---|---|---|
| TimeSqueeze $_{\text{base}}$ | 12 | 12 | 8 | 2 | 384 | 1536 | 192 | 50M | 113M |
| TimeSqueeze $_{\text{large}}$ | 12 | 12 | 8 | 2 | 768 | 3072 | 384 | 200M | 453M |

(b) Transformer backbone.

# B    DOWNSAMPLING OF PRETRAINING DATASET

The original Time-300B dataset is heavily skewed by the Nature domain, which contributed to more than 90% of the dataset, as shown in Table 4.

Table 4: Key statistics of the pre-training dataset Time-300B from various domains.

|  | Energy | Finance | Healthcare | Nature | Sales | Synthetic | Transport | Web | Other | Total |
|---|---|---|---|---|---|---|---|---|---|---|
| # Seqs. | 2,875,335 | 1,715 | 1,752 | 31,621,183 | 110,210 | 11,968,625 | 622,414 | 972,158 | 40,265 | 48,220,929 |
| # Obs. | 15.981 B | 413.696 K | 471.040 K | 279.724 B | 26.382 M | 9.222 B | 2.130 B | 1.804 B | 20.32 M | 309.09 B |
| Percent % | 5.17% | 0.0001% | 0.0001% | 90.50% | 0.008% | 2.98% | 0.69% | 0.58% | 0.006% | 100% |

And within the Nature domain, the 3 largest domains datasets contribute the most, as seen in Table 5.

Table 5: Key properties of Nature dataset from Time-300B..

| Dataset | Domain | Freq. | # Time Series | # Obs. | Source |
|---|---|---|---|---|---|
| Weatherbench (Hourly) | Nature | H | 3,984,029 | 74,630,250,518 | (Rasp et al., 2020) |
| Weatherbench (Daily) | Nature | D | 301,229 | 3,223,513,345 | (Rasp et al., 2020) |
| Weatherbench (Weekly) | Nature | W | 226,533 | 462,956,049 | (Rasp et al., 2020) |
| Beijing Air Quality | Nature | H | 4,262 | 2,932,657 | (Chen, 2019) |
| China Air Quality | Nature | H | 17,686 | 4,217,605 | Zheng et al. (2015) |
| CMIP6 | Nature | 6H | 14,327,808 | 104,592,998,400 | Nguyen et al. (2023) |
| ERA5 | Nature | H | 11,940,789 | 93,768,721,472 | Nguyen et al. (2023) |
| Oikolab Weather | Nature | H | 309 | 615,574 | (Godahewa et al., 2021) |
| Saugeen | Nature | D | 38 | 17,311 | (Godahewa et al., 2021) |
| Subseasonal | Nature | D | 17,604 | 51,968,498 | (Mouatadid et al., 2023) |
| Subseasonal Precipitation | Nature | D | 13,467 | 4,830,284 | (Mouatadid et al., 2023) |
| Sunspot | Nature | D | 19 | 45,312 | (Godahewa et al., 2021) |
| Temperature Rain | Nature | D | 13,226 | 3,368,098 | (Godahewa et al., 2021) |
| Weather | Nature | D | 9,525 | 26,036,234 | (Ansari et al., 2024) |

In order to reduce the bias from these 3 datasets, we downsample the top 3 datasets by 30% at random during pretraining, bringing down the total number of samples in the pretraining dataset from 309B to $\approx$120B.

## C  TRAINING TOKENS VS PERFORMANCE

Figure 4 demonstrates that TimeSqueeze exhibits favorable scaling behavior, with performance consistently improving as the training budget increases from 10B to 50B tokens. This scaling trend aligns with observations in (Shi et al., 2024), indicating that TimeSqueeze can effectively leverage larger datasets and computational resources. The consistent performance gains across different training scales suggest that TimeSqueeze exhibits similar scaling behavior to Time-MoE but with significantly improved data and compute efficiency, positioning it as a promising candidate for even larger-scale pretraining regimes.

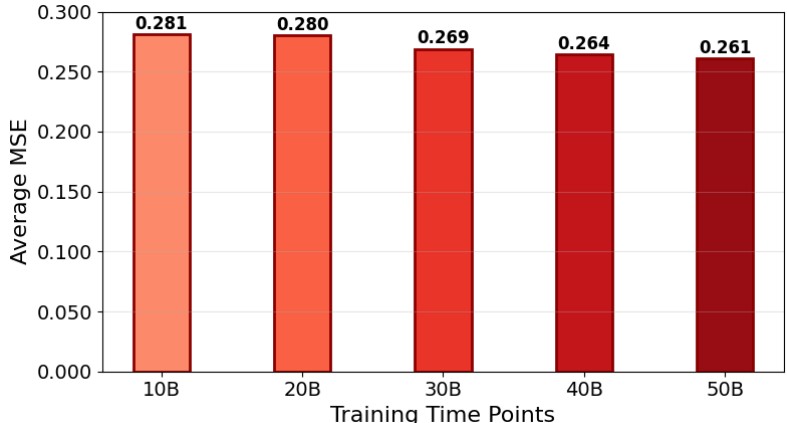

Figure 4: Performance scaling with training data size: Average MSE for 96-horizon forecasting across five benchmarks shows consistent improvement with increased training tokens.

## D  COMPRESSION RATE VS PERFORMANCE

For the main results, we choose a moderate compression rate of $4\times$. We now compare the performance against two more variants of TimeSqueeze $_{\text{base}}$ trained with a target compression rate of $6\times$ and $8\times$, by adjusting the threshold factor to $0.4$ and $0.45$ respectively. And we plot the average MSE across the five datasets for prediction horizon 96. As expected, while the computational efficiency increases with higher compression, the performance also drops noticeably. techniques such as heirarchical compression

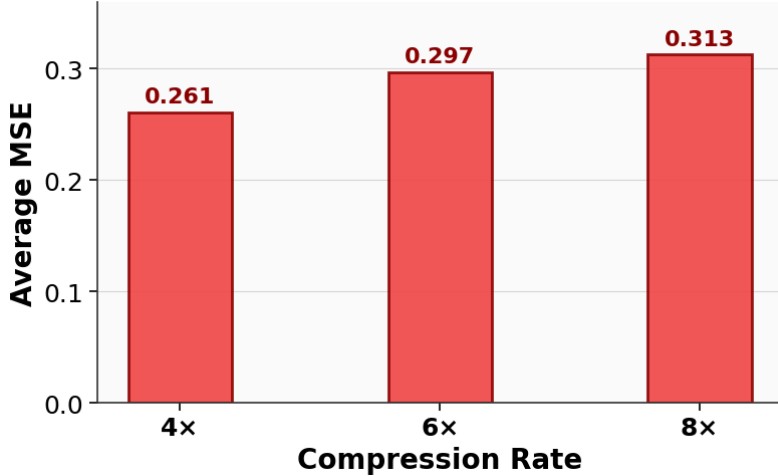

Figure 5: Performance scaling with training data size: Average MSE for 96-horizon forecasting across five benchmarks shows consistent improvement with increased training tokens.

## E    PERFORMANCE FOR A FIXED CONTEXT LENGTH

TimeSqueeze offers two key advantages over Time-MoE: First is the reduced token count to the Transformer backbone through dynamic compression. Further, TimeSqueeze also improves forecasting capability over longer horizons using shorter historical contexts, compared to point embedding models.

Our analysis demonstrates that for a fixed context length, TimeSqueeze significantly outperforms the point embedding baseline Time-MoE when predicting long-horizon forecasts. Figure 6 shows that for a given context length of 512, TimeSqueeze achieves a superior forecasting accuracy for the a horizon of 336. This improvement stems from our adaptive patching mechanism, which enables the model to extract more informative temporal patterns from limited historical data.

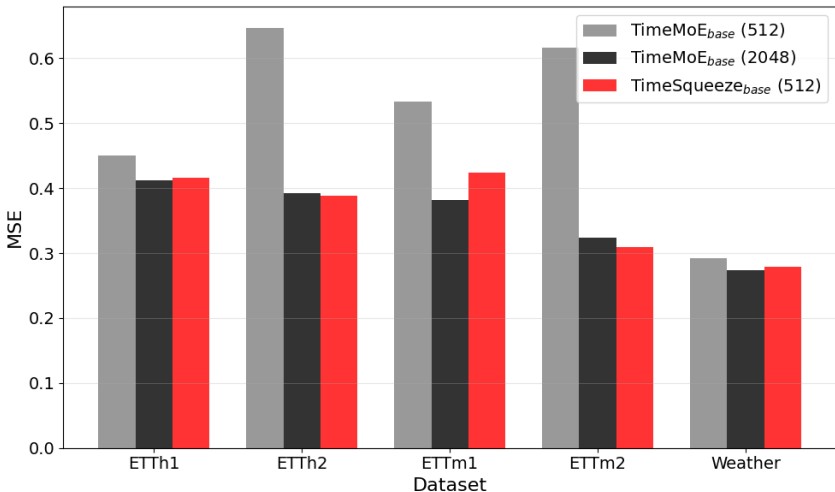

Figure 6: Performance comparison between TimeSqueeze and Time-MoE for a given context length for prediction horizon 336. TimeSqueeze noticeably outperforms the point-embedding baseline when the available context is limited.

## F    ADDITIONAL ABLATION RESULTS

Table 6 contains the full et of results for the ablation studies presented in Section 4.4.

Table 6: Ablation study on zero-shot forecasting performance for prediction horizon 96.

| Model / Variation | ETTh1 | | ETTh2 | | ETTm1 | | ETTm2 | | Weather | | Average | |
|---|---|---|---|---|---|---|---|---|---|---|---|---|
| | MSE | MAE | MSE | MAE | MSE | MAE | MSE | MAE | MSE | MAE | MSE | MAE |
| TimeSqueeze $_{base}$ | 0.357 | 0.384 | 0.281 | 0.336 | 0.311 | 0.343 | 0.181 | 0.270 | 0.166 | 0.216 | 0.259 | 0.310 |
| Time-MoE$_{base}$ | 0.357 | 0.381 | 0.305 | 0.359 | 0.338 | 0.368 | 0.201 | 0.291 | 0.160 | 0.214 | 0.272 (+5.0%) | 0.323 (+4.2%) |
| TimeSqueeze w/ fixed patching | 0.373 | 0.396 | 0.455 | 0.448 | 0.359 | 0.382 | 0.335 | 0.380 | 0.178 | 0.232 | 0.340 (+31.3%) | 0.368 (+18.7%) |
| TimeSqueeze w/ linear patching (no SSM) | 0.379 | 0.401 | 0.481 | 0.463 | 0.370 | 0.375 | 0.375 | 0.402 | 0.158 | 0.174 | 0.353 (+36.3%) | 0.363 (+17.1%) |
| TimeSqueeze w/o fine-grained features | 0.366 | 0.388 | 0.277 | 0.342 | 0.339 | 0.362 | 0.187 | 0.283 | 0.169 | 0.218 | 0.268 (+3.5%) | 0.319 (+2.9%) |
| TimeSqueeze w/o original pos. IDs | 0.375 | 0.393 | 0.291 | 0.358 | 0.346 | 0.363 | 0.191 | 0.293 | 0.169 | 0.219 | 0.274 (+5.8%) | 0.325 (+4.8%) |

### F.1    INFERENCE CONTEXT LENGTH VS PERFORMANCE

While longer context lengths generally provide more historical information for forecasting, the relationship between context length and performance is not monotonic. We investigate the effect of varying inference context lengths on forecasting accuracy by evaluating TimeSqueeze with context lengths ranging from 96 to 1536 tokens while keeping all other hyperparameters fixed.

Figure 7 reveals that performance initially improves as context length increases from 96 to 1536, reaching optimal performance around 512. However, further increasing the context length beyond

this range leads to marginal performance degradation. This suggests that while additional historical context can be beneficial up to a certain point, excessively long contexts may introduce noise or make it harder for the model to focus on the most relevant patterns.

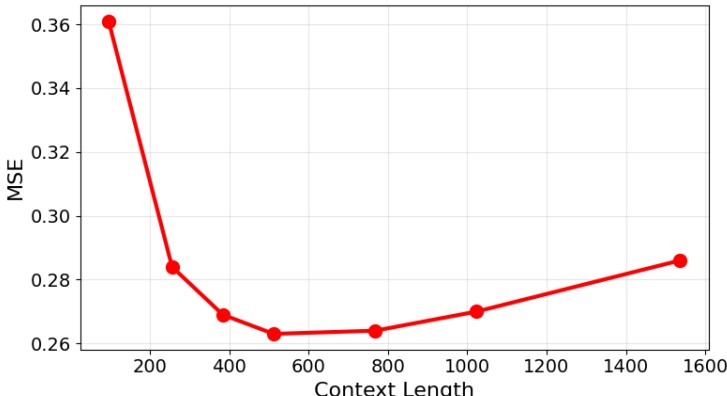

Figure 7: Forecasting performance improves with context length up to 512-800 tokens, then plateaus or slightly degrades.

## G    VISUALIZATION OF PATCHING

We provide the visualization of dynamic patches computed for an example segment of 128 samples from each of the evaluation datasets in Figures 8, 9, and 10. As we can see, weather dataset has slower variation in data resulting in larger patch sizes, whereas ETTm data has several regions with rapidly varying signal, resulting in much smaller patch sizes.

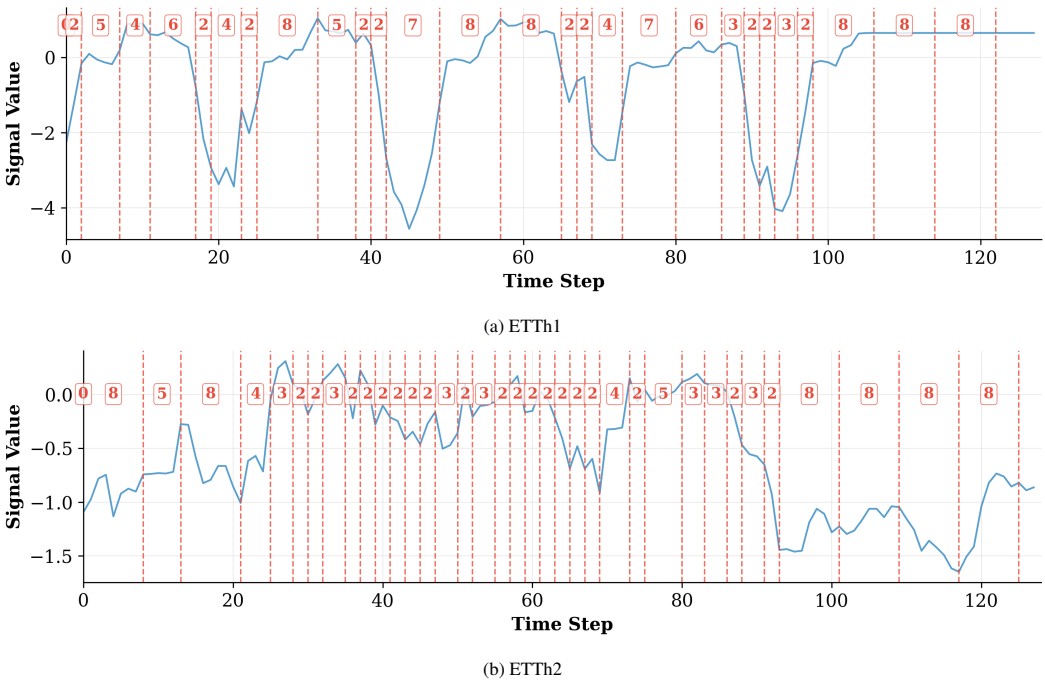

Figure 8: Example dynamic patch boundaries for ETTh1 and ETTh2 datasets.

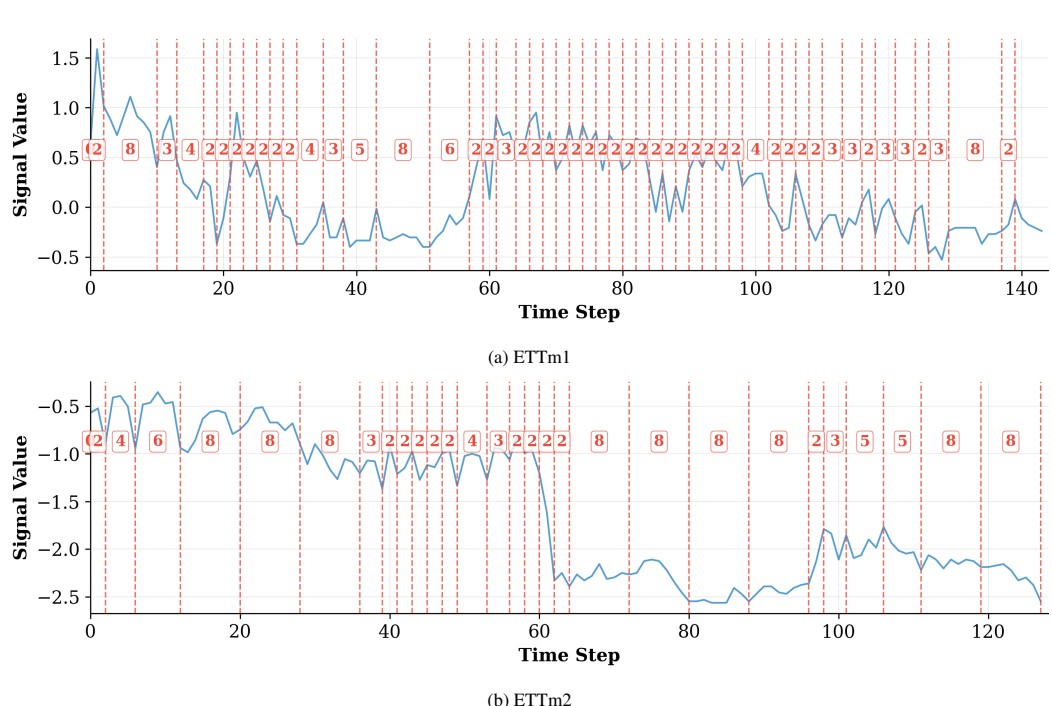

Figure 9: Example dynamic patch boundaries for ETTm1 and ETTm2 datasets.

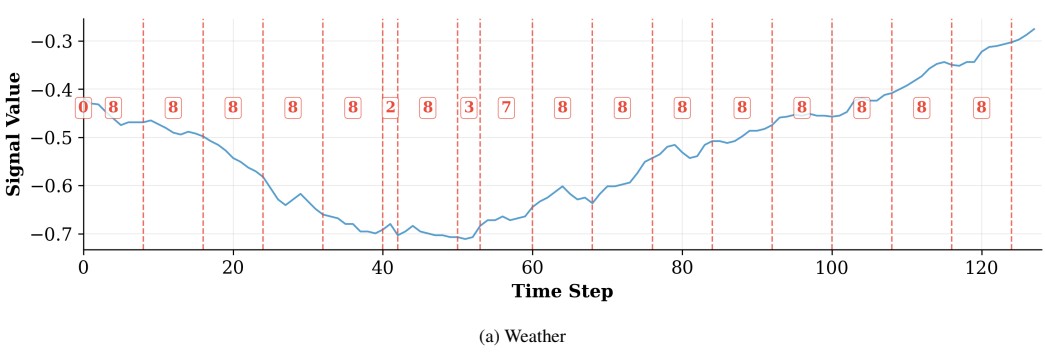

Figure 10: Example dynamic patch boundaries for Weather dataset.

# H PATCH DISTRIBUTION

We provide the visualization of patch size distributions for each of the eval datasets in Figures 11. As we can see, weather dataset has slower variation in data resulting in larger avg patch size, whereas ETTm2 data has several regions with rapidly varying signal, resulting in much smaller patch sizes.

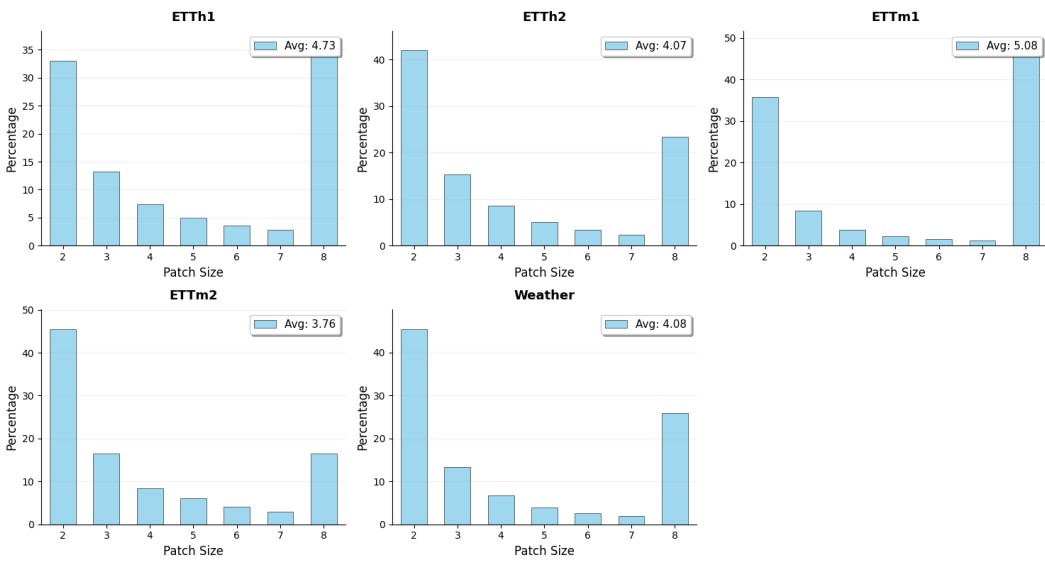

Figure 11: Distribution of patch sizes across eval datasets.

# I VISUALIZATION OF FORECASTS

We provide the visualization of forecasting results for an example segment of 128 samples from each of the evaluation datasets in Figures 12, 13, and 14. As observed, the Weather dataset exhibits relatively smooth and slowly varying dynamics, making forecasts easier to capture, whereas the ETTm datasets contain regions with rapid fluctuations, which pose greater challenges for accurate prediction.

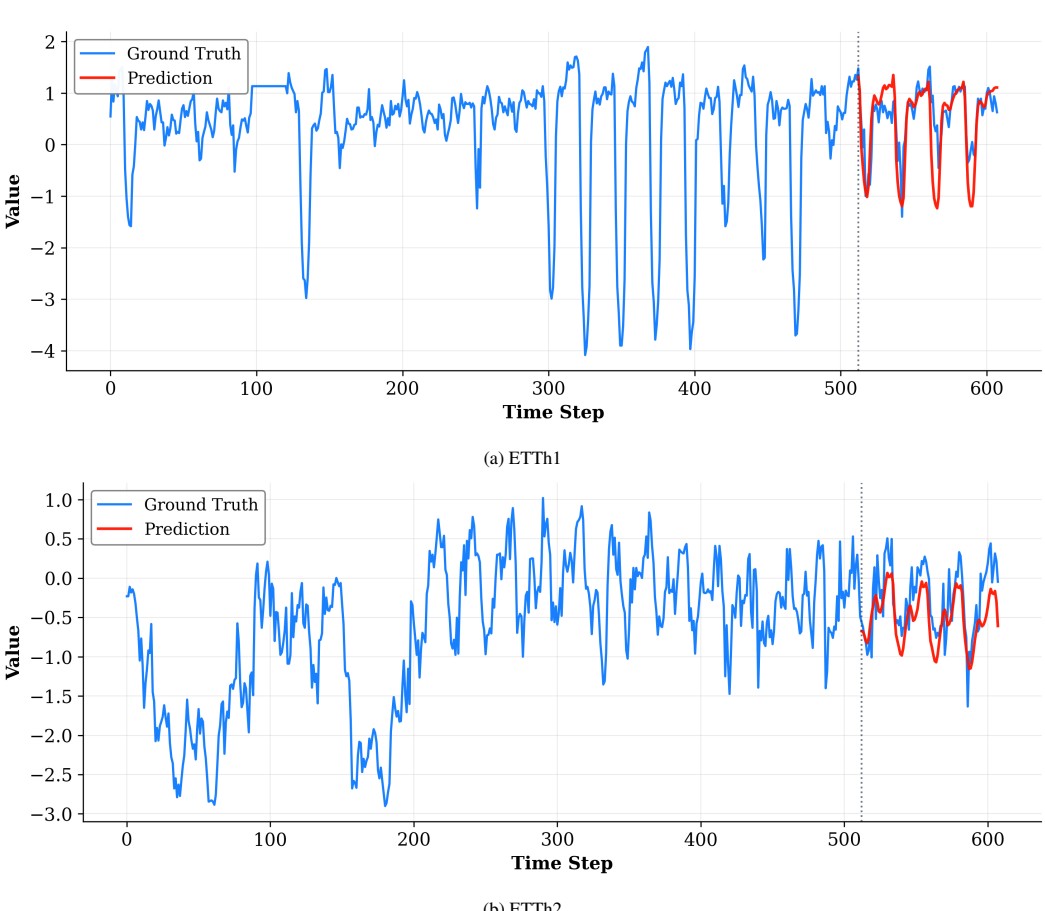

(a) ETTh1

(b) ETTh2

Figure 12: Forecasting results on ETTh1 and ETTh2 datasets.

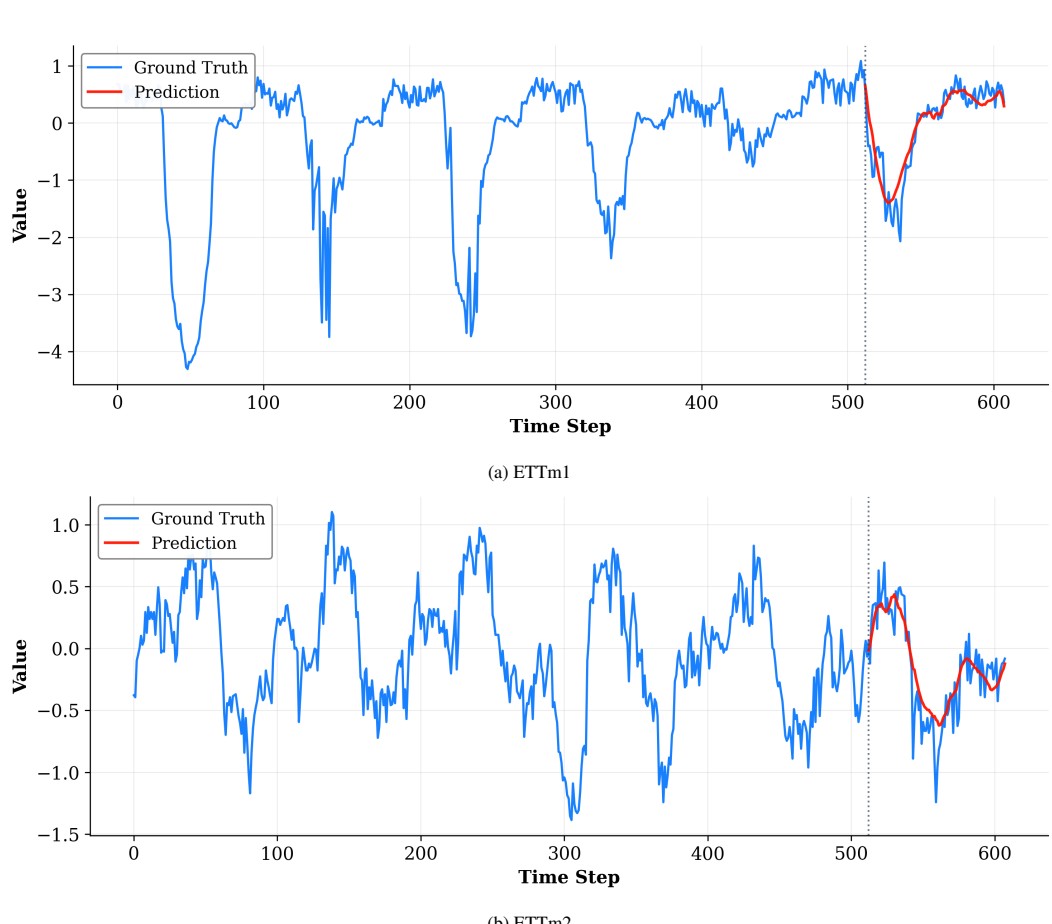

(a) ETTm1

(b) ETTm2

Figure 13: Forecasting results on ETTm1 and ETTm2 datasets.

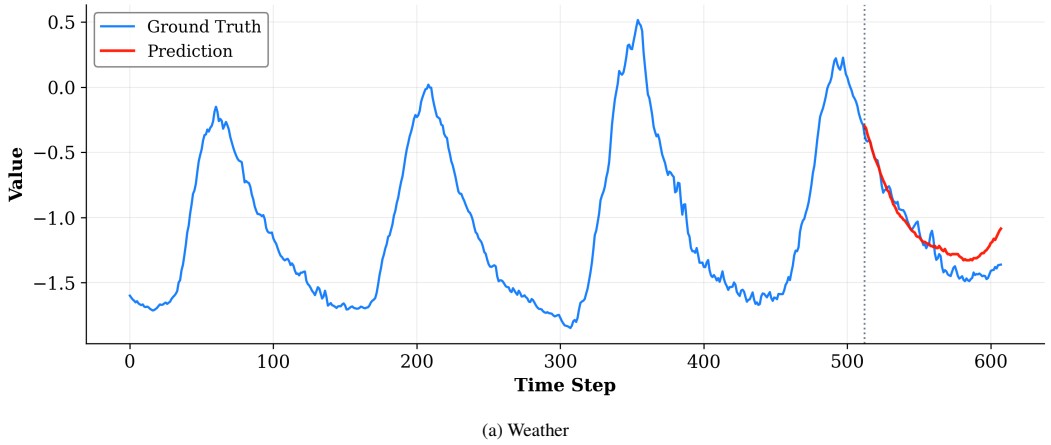

(a) Weather

Figure 14: Forecasting results on the Weather dataset.

