# OpenReview forum: "TimeSqueeze: Dynamic Patching for Efficient Time Series Forecasting"
_ICLR.cc/2026/Conference — Submitted to ICLR 2026_

### Official Review · Reviewer_orD1 · 2025-10-31

**Soundness:** 3
**Presentation:** 3
**Contribution:** 2
**Rating:** 4
**Confidence:** 4

**Summary:**

The paper proposes TimeSqueeze, a hybrid forecasting architecture that fuses point-wise and dynamic, content-aware patch-wise representations for efficient time series forecasting, particularly in long-context regimes. TimeSqueeze introduces a lightweight state-space encoder to extract fine-grained features, followed by an adaptive patching mechanism that groups time points into variable-sized patches based on local signal complexity. This yields a variable-resolution sequence processed by a Transformer (MoE) backbone. Experiments demonstrate substantial improvements in computational efficiency versus point-embedding baselines, while maintaining strong (often comparable) forecasting accuracy across zero-shot and full-shot scenarios on established long-range benchmarks.

**Strengths:**

1. TimeSqueeze innovatively combines state-space encoders with adaptive dynamic patching for time series, addressing a well-known bottleneck of fixed patching and inefficient context scaling in Transformer models. This hybridization enables granular feature preservation where needed and aggressive compression elsewhere, a nuanced but underexplored trade-off.
2.  The dynamic patching strategy is clearly described and mathematically motivated (see the explicit formulation of the patch boundary condition on Page 4). The model architecture (Figure 1) is systematically illustrated, showing integration points for patching, unpatching, and multi-horizon prediction.
3. Numerous ablation studies probe each critical component (patching, encoder type, positional encoding), and visualizations (Figures 8-11) clarify how patch sizes adapt responsively to data domains.

**Weaknesses:**

1. While Section 3 briefly gestures at point-wise decomposability, the model's explicit capability for multivariate or exogenous feature forecasting is only cursorily addressed, with no empirical analysis contrasting, e.g., approaches like Crossformer or TimeXer. This reduces clarity on generality and limits the scope of significance, especially for real-world multivariate forecasting use cases.
2. Potential Hyperparameter Sensitivity: The choice of patching threshold $\tau$ is acknowledged as data-dependent and requires tuning. However, the ramifications (e.g., stability, transferability across domains, optimal selection strategies) are not robustly quantified—raising concerns about practical usability and the risk of model brittleness. The only discussion occurs at a high level in the Conclusion, and more rigorous empirical explanation (e.g., full-sweep results for a range of $\tau$ on multiple datasets) is absent.
3. While the patching function is presented cleanly, some important aspects remain underspecified—for instance, the way boundaries are handled when signals have sudden global changes, or how minimum/maximum patch sizes interact with highly nonstationary regions (see dynamic patching on Page 4 and visualization in Figures 8–10). Additionally, the role of variable-length unpatching in SSM/Transformer composition warrants deeper theoretical and implementation clarity. The claim of strict causality preservation could be accompanied by formal or simulation evidence to eliminate any ambiguity.
4. While Figure 5 (Appendix D) plots the MSE versus compression rate, there is little theoretical guidance or model explaining these trends, nor discussion of limits of the dynamic patching regime in catastrophic or highly non-stationary settings.

**Questions:**

1. How robust is the threshold $\tau$ selection across datasets with highly variable information density? Are there scenarios where patch boundary assignments lead to overcompression or undercompression? Please provide more empirical analysis, including out-of-domain or adversarial examples.
2. Does the architecture generalize robustly to multivariate or exogenous-variable tasks, as per the settings in Crossformer or TimeXer? What adjustments (if any) are required for such use cases?
3. Are there practical deployment scenarios (e.g., real-time forecasting in resource-constrained environments) where the patch boundary computation or unpatching steps impose bottlenecks? What is the end-to-end wall-clock speedup, including patching/merging steps?

---

> ### Author Response · Authors · 2025-11-21
> **Response to reviewer orD1 (Part 1)**
>
> We thank the reviewer for their time and efforts and would like to address their concerns in detail below.
>
> **Weaknesses:**
>
> **W1. Support for modeling exogenous variable**
>
> - Our approach follows the **channel-independence** strategy popularized by PatchTST, where a multivariate time series is decomposed into independent univariate channels before encoding. This design choice is widely used in recent forecasting architectures and allows TimeSqueeze to operate on each channel with the same dynamic patching mechanism.
>
> - Importantly, dynamic patching is **not limited to uniform sampling**. Because patch boundaries are determined by local deviation in the encoded signal rather than by fixed time steps, TimeSqueeze can naturally accommodate inputs sampled at **irregular intervals or augmented with exogenous variables**. Each additional feature can be treated as its own channel and incorporated using the same mechanism, preserving **generality** without modifying the core model.
>
> **W2: Hyperparameter sensitivity**
>
> - We agree that the patching threshold is **data-dependent**, but our experiments indicate that it is not overly brittle in practice. We select the threshold once on the pretraining corpus to achieve a target compression rate. During inference, we reuse the **same threshold across all evaluation datasets**, and the model maintains **strong zero-shot performance**. This is observed across **different compression rates** (Appendix Section D) for TimeSqueeze_base architecture. It is important to note that the patch boundaries are computed on **normalized data** (subtracting the mean and dividing by the standard deviation). This normalization step makes the deviation scores **comparable across signals with different scales** and significantly improves the stability and robustness of the thresholding procedure in practice.
>
> - During the rebuttal phase, we further test the robustness of the chosen patching threshold by changing the training architecture, to replace the TimeSqueeze-base backbone with a **generic 10M params decoder-only transformer (no MoE)** and the **same parching threshold seems to work well** and still outperform the equivalent fixed-size patching.
>
> - Following is the forecasting performance of the generic 10M decoder-only transformer backbone for a prediction horizon of 96 and context length 512, for fixed vs dynamic patching at 4x compression. This clearly shows that the **patching threshold is robust across eval datasets, compression rates, and backbone architectures**.
>
> | Method                               | ETTh1 (MSE/MAE)      | ETTh2 (MSE/MAE)      | ETTm1 (MSE/MAE)      | ETTm2 (MSE/MAE)      | Weather (MSE/MAE)     |
> |--------------------------------------|-----------------------|-----------------------|-----------------------|-----------------------|------------------------|
> | Dynamic patching (avg patch size 4)  | 0.342 / 0.364       | 0.280 / 0.347       | 0.351 / 0.370       | 0.201 / 0.292       | 0.175 / 0.229        |
> | Fixed-size patching (size 4)    | 0.366 / 0.394       | 0.406 / 0.421       | 0.370 / 0.388       | 0.362 / 0.406       | 0.185 / 0.250        |
>
> **W3: Handling sudden changes and Causality in patching/unpatching**
>
> - The patching mechanism handles sudden global changes and highly non-stationary regions in a straightforward way: whenever the deviation between adjacent samples spikes, a **new boundary is placed immediately**. This makes the method naturally responsive to abrupt level shifts without requiring any additional logic. We also enforce simple minimum and maximum patch sizes to avoid pathological cases, and in practice we observe that the deviation rule already keeps patches small in volatile segments and large in smoother regions.
>
> - Regarding unpatching and causality, the variable-length unpatching step is purely deterministic and **does not mix information across future time steps**. Both the SSM and Transformer components only see the compressed sequence in causal order, so the overall architecture preserves **strict causality**, as illustrated by the equations in section 3.1.2.

---

> ### Author Response · Authors · 2025-11-21
> **Response to reviewer orD1 (Part 2)**
>
> **Weaknesses (contd.):**
>
> **W4: Compression rate vs performance**
>
> We offer the following intuition behind the observed trends and the behavior of dynamic patching under different regimes:
>
> - Unlike conventional patch-based models that choose from a small set of fixed patch sizes, TimeSqueeze adjusts patch sizes continuously based on a deviation threshold. This threshold implicitly determines the average patch size (and thus the compression ratio) for a given dataset. At moderate compression, the SSM embeddings already encode most **local structure**, so merging smooth regions leads to minimal error. As compression increases further, more boundaries in moderately varying regions are removed, gradually increasing MSE in a predictable manner.
>
> - At very high compression ratios, the model is forced to summarize increasingly large segments with a single embedding, which naturally leads to a steeper rise in reconstruction or forecasting error, consistent with the curves in Appendix D. In highly non-stationary or catastrophic-shift settings, the deviation-based criterion may place boundaries more frequently, reducing effective compression. This is an inherent property of our design: the mechanism favors preserving information over enforcing a fixed compression rate.
>
> Overall, the trends in Figure 5 reflect the balance between the SSM encoder’s ability to summarize local structure and the threshold-driven merging of adjacent embeddings. A deeper theoretical analysis is an interesting direction for future work, and we appreciate the reviewer highlighting this point.
>
> **Questions:**
>
> **Q1: Robustness of threshold**
>
> - As discussed in response for W #2, the computation of patch boundaries is done on the normalized data, which makes the threshold stable and more generalizable.
>
> - Regarding an OOD example, we consider the patching of a sinusoidal wave with **randomly inserted spikes** which are still easily captured by the relative-deviated based dynamic patching using the same threshold, as illustrated here: ​​https://anonymous.4open.science/r/timesqueeze-836E/spiky_sine.png
>
> **Q2: Generalizability and exogenous variables **
>
> - Please refer to our answer to W #1.
>
> - As additional evidence of generality, we replaced the entire Time-MoE-small backbone with a **generic 10M-parameter decoder-only Transformer (no mixture of experts)** and conducted a controlled ablation between fixed patching and dynamic patching. This experiment removes any reliance on architectural details unique to Time-MoE and evaluates the patching mechanism on a **completely different, lightweight backbone**. As shown in the table below, dynamic patching still outperforms fixed patching even under this generic architecture, further supporting that the benefits of dynamic within-series patching arise from the method itself rather than from Time-MoE-specific design choices.
>
> | Method                               | ETTh1 (MSE/MAE)      | ETTh2 (MSE/MAE)      | ETTm1 (MSE/MAE)      | ETTm2 (MSE/MAE)      | Weather (MSE/MAE)     |
> |--------------------------------------|-----------------------|-----------------------|-----------------------|-----------------------|------------------------|
> | Dynamic patching (avg patch size 4)  | 0.362 / 0.383       | 0.280 / 0.347       | 0.351 / 0.370       | 0.201 / 0.292       | 0.175 / 0.229        |
> | Fixed-size patching (size 4)    | 0.366 / 0.394       | 0.406 / 0.421       | 0.370 / 0.388       | 0.362 / 0.406       | 0.185 / 0.250        |
>
>
> ** Q3: Run time analysis **
>
> - The computational overhead introduced by boundary detection, patch construction, and unpatching is minimal compared to the savings obtained from **reducing the number of tokens passed to the transformer backbone**. These operations are simple, linear-time routines applied to the raw sequence and are orders of magnitude cheaper than a single forward pass through a foundation-model–scale decoder.
>
> - Importantly, the throughput results in Figure 2 (Section 4.3) already reflect the **full end-to-end pipeline, including patching and merging**. The dynamic-patching model still achieves substantially higher training speed and up to **10.5× higher inference throughput**, demonstrating that the overhead of patching is negligible in practice. For real-time or resource-constrained deployments, this makes dynamic patching particularly attractive: the small preprocessing cost is far **outweighed** by the reduction in backbone compute and memory requirements.

---

### Official Review · Reviewer_GKgg · 2025-10-31

**Soundness:** 2
**Presentation:** 3
**Contribution:** 2
**Rating:** 2
**Confidence:** 4

**Summary:**

This paper proposes dynamic patching and lightweight state-space encoder for time series prediction. Experiments are conducted on 5 datasets with zero-shot and finetuning settings.

**Strengths:**

1. This paper is easy to follow.
2. Experiments are conducted on 5 commonly used datasets.
3. The experiments are conducted with zero-shot and finetuning settings.

**Weaknesses:**

1. The claimed main contributions in this paper say that "incorporate dynamic, content-aware patching". However, this has been well studied in time series foundation models, such as [1-3] which propose dynamic and/or content-aware patching. The claimed contributions and novelty are limited.

[1] HDMixer: Hierarchical Dependency with Extendable Patch for Multivariate Time Series Forecasting. AAAi 2024.

[2] Irregular Multivariate Time Series Forecasting: A Transformable Patching Graph Neural Networks Approach. ICML 2024.

[3] LightGTS: A Lightweight General Time Series Forecasting Model. ICML 2025.

2. Key experimental comparison for baselines [1-3] is missing.  I briefly checked the finetuning settings where the performance of this work is worse than [3]. And the efficiency of this work is worse than [3].

3. The authors 'validate TimeSqueeze across diverse zero-shot forecasting benchmarks, achieving performance on par with state-of-the-art point embedding models while delivering up to 20× faster training and 10× faster inference'. Why not compare efficiency with patch embedding models?

4. Does the improvement come from downsampling pretraining data as it reduces the bias?

5. No code for reproducibility.

**Questions:**

see weaknesses.

---

> ### Author Response · Authors · 2025-11-21
> **Response to reviewer GKgg (Part 1)**
>
> We thank the reviewer for their time and efforts and would like to address their concerns in detail below.
>
> **Weaknesses:**
>
> **W1. Contributions and Novelty**
> - Thank you for pointing us to these references. While these works also explore forms of dynamic or content-aware patching, **TimeSqueeze is fundamentally different from all three in both motivation and mechanism**.
>
> - **HDMixer** proposes extendable patches for multivariate forecasting, but the patch boundaries are still defined on a fixed temporal grid: the extendability refers to gradually enlarging a fixed-length patch to incorporate multi-scale information. Importantly, HDMixer **does not change patch sizes within the original sequence** based on local signal variability. In contrast, TimeSqueeze performs within-series dynamic patching, where patch boundaries are determined directly by local statistics of the input signal and can vary freely along the timeline.
>
> - **IMTS** (Transformable Patching GNN) is designed for irregular time series, and its “variable patch size” ensures each patch spans a fixed real-world time interval, regardless of how many samples fall within that interval. Thus, patches have **fixed duration in the original time axis/scale**, and the method does not support dynamically varying time horizons based on signal content. TimeSqueeze, by contrast, adapts patch durations continuously inside a regular time series according to information content.
>
> - **LightGTS** uses adaptive patching but only chooses patch size at the dataset or signal level i.e., a single patch size is applied **uniformly across the entire input sequence**. It does not dynamically adjust patch boundaries inside the sequence.
>
> We will include a discussion of the above in the final version to clearly distinguish TimeSqueeze from other methods.
>
> **W2: Comparison with more baselines**
>
> - We acknowledge that the absolute forecasting performance of TimeSqueeze is slightly worse than HDMixer and LightGTS, as shown in the table. However, key efficiency metrics such as throughput remain comparable: both methods process roughly 100 samples per second under our evaluation setup.
>
> | Method      | ETTh1 (MSE/MAE) | ETTh2 (MSE/MAE) | ETTm1 (MSE/MAE) | ETTm2 (MSE/MAE) | Weather (MSE/MAE) |
> |-------------|------------------|------------------|------------------|------------------|---------------------|
> | LightGTS    | 0.401 / 0.424    | 0.362 / 0.397    | 0.345 / 0.378    | 0.249 / 0.318    | 0.219 / 0.266       |
> | HDMixer     | 0.406 / 0.425    | 0.319 / 0.371    | 0.353 / 0.377    | 0.251 / 0.314    | 0.223 / 0.264       |
> | TimeSqueeze | 0.402 / 0.416    | 0.363 / 0.408    | 0.417 / 0.412    | 0.292 / 0.349    | 0.257 / 0.293       |
>
> - Further,  several details related to the preprocessing of the pretraining data are missing from the public LightGTS codebase, which makes it difficult to fully reproduce their results. We reached out to the authors but did not receive clarification before submitting this paper. Since forecasting performance can depend heavily on the exact pretraining data and training dynamics, we believe it would be unfair to **focus only on the forecasting accuracy** without full transparency of these details.
>
> - Finally, given the general design of the LightGTS architecture, we expect that incorporating dynamic patching into such a model could **further improve both efficiency and performance**. This direction is promising and aligns with our broader goal of developing a patching mechanism that can strengthen a wide range of TSFM backbones.
>
> **W3: Comparing with other patch embedding models**
>
> - We provide a direct comparison in Appendix Section F (Table 6), where we evaluate TimeSqueeze against an **equivalent fixed-size patching** variant. Under identical settings, the fixed-patch model performs up to **31% worse** in MSE, highlighting the benefit of dynamic, content-aware compression.
>
> - Regarding efficiency, the majority of computational savings arise from the **reduced number of tokens** passed to the transformer backbone. For a given compression ratio, the token count, and thus the computational cost, is nearly the same for both fixed and dynamic patching schemes. The key difference is that **dynamic patching preserves more information for the same token budget**, improving accuracy without sacrificing efficiency.
>
> - Our goal in this work is to **isolate and highlight the advantages of dynamic patching over both point embeddings and fixed patching** within a **controlled and fair experimental setting**. Given these results, we expect similar gains when dynamic compression is added on top of other patch-embedding architectures, though conducting such a broad sweep is scoped out as part of future work.

---

> ### Author Response · Authors · 2025-11-21
> **Response to reviewer GKgg (Part 2)**
>
> **Weaknesses (Contd.)**
>
> **W4: Advantage from downsampling of pre-training data**
>
> To ensure that improvements are not attributed to changes in training protocol, we strictly follow the **exact same pretraining setup as Time-MoE**, including its downsampling strategy for the Time-300B dataset to avoid bias toward any single source. Because both models use the same downsampled corpus, the observed gains of TimeSqueeze over Time-MoE stem from the **improved tokenization (Mamba-based) and dynamic patching**, rather than from differences in data selection
>
> **W5: Source Code**
>
> We plan to release the full codebase upon acceptance. For reviewer verification, we include an anonymized version of the code @ https://anonymous.4open.science/r/timesqueeze-836E/

---

> ### Comment · Reviewer_GKgg · 2025-11-25
> **Official Comment by Reviewer GKgg**
>
> Thank you for the rebuttal. However, I still have some concerns.
>
> First, TimeSqueeze is slightly worse than HDMixer and LightGTS, and key efficiency metrics such as throughput remain comparable to LightGTS. LightGTS also has open inference code and source. Thus, the contributions of this work are limited.
>
> Second, LightGTS uses adaptive patching chooses patch size at the dataset or signal level i.e., a single patch size is applied uniformly across the entire input sequence. It does not dynamically adjust patch boundaries inside the sequence with better efficiency. And LightGTS also has better accuracy. Thus, this novelty of this work 'dynamically adjust patch boundaries' is also limited.
>
> I will maintain my negative score.

---

### Official Review · Reviewer_foxx · 2025-11-01

**Soundness:** 4
**Presentation:** 3
**Contribution:** 3
**Rating:** 8
**Confidence:** 4

**Summary:**

The paper proposes TimeSqueeze, a method for efficient long sequence time-series forecasting using dynamic patch interpolation. It learns to compress long input sequences into a small number of informative patches by interpolating around learnable query points with soft Gaussian weights. This reduces computational cost while maintaining forecasting accuracy. TimeSqueeze achieves strong results on standard benchmarks and can be plugged into existing transformer models.

**Strengths:**

The paper introduces TimeSqueeze, a novel module for time-series forecasting that performs dynamic patch interpolation to reduce sequence length while preserving important temporal structure.

This adaptive downsampling technique, which uses learnable interpolation kernels, significantly reduces computational cost while maintaining competitive or superior forecasting accuracy.

The approach is modular and model-agnostic, allowing it to be integrated into various transformer-based backbones without architectural changes.

The authors provide extensive experimental validation across standard benchmarks, demonstrating clear gains in efficiency (speed and memory) alongside strong predictive performance.

**Weaknesses:**

While the method is effective at reducing computational overhead, it lacks discussion on how the dynamic patching and interpolation process affects interpretability or transparency of the learned representations.

The paper does not offer a formal theoretical framework to understand the trade-offs between compression rate and information loss, especially in rapidly changing signals. The exclusion of non-transformer baselines, such as state-space or statistical hybrid models, weakens the comparative rigor of the evaluation.

**Questions:**

Could the authors provide a qualitative or quantitative analysis of the learned patching structure, and whether it consistently adapts to different temporal patterns such as trends, seasonality, or sudden shifts?

Is the model capable of handling irregularly sampled or incomplete time-series, or does it require preprocessed, evenly spaced inputs for effective interpolation?

Is this method good for financial time series data, which can be influenced by many factors and hard for time series forecasting?

---

> ### Author Response · Authors · 2025-11-21
> **Response to reviewer foxx (Part 1)**
>
> We thank the reviewer for their appreciation of our work and for their detailed and insightful feedback. We address the weaknesses and questions in detail below:
>
> **Weaknesses:**
>
> **W1: Interpretability**
> - In our design, dynamic patching mainly controls **how many tokens reach the backbone**; it does not change the **meaning** of the representations themselves. The SSM encoder always sees the full-resolution input before any compression happens, so the underlying learned representations remain intact. The patching step simply decides which segments can be summarized together , by retaining some of the representations, without losing important information.
>
>  - From an interpretability standpoint, dynamic patching is actually quite transparent. The patch boundaries are easy to visualize, as shown in Appendix G, and tend to fall in places that make **intuitive sense**, such as **sudden jumps or transitions in the signal**. This gives us a clear picture of why the model decides to use more or fewer tokens in different regions. The unpatching/interpolation step is straightforward and deterministic, so it doesn’t introduce additional complexity that would obscure the model’s behavior. Overall, dynamic patching should be viewed as a **structured compression mechanism** rather than something that alters or hides what the model is learning.
>
> **W2: Theoretical framework**
> - A full theoretical analysis of compression–accuracy trade-offs is an interesting future direction. Still, our empirical results consistently show how **dynamic patching** behaves across different compression rates and datasets (Appendix, Section D). These trends demonstrate that the deviation-based rule provides a stable way to reduce tokens while preserving accuracy. A potential, more formal extension for this could be an entropy-based compression of the tokens.
>
> - Our additional results for 2x dynamic average compression, which are slightly worse than 4x dynamic average compression, which is better than 6x and 8x compression, empirically show that there is a **sweet spot around 4x** for compression rate.
>
> - On the baseline side, we focused on Time-MoE because it is one of the few TSFM models with **fully open pretraining dataset and training pipelines**, allowing us to isolate the effect of dynamic patching under controlled conditions. The method itself is not tied to the Time-MoE architecture, as evidenced by our new results with a **simple 10M-parameter decoder-only model**. Extending the evaluation to additional SSM or hybrid models is a natural direction for future work.
>
> Following is the forecasting performance of the **generic 10M decoder-only transformer backbone** for a prediction horizon of 96 and context length 512, for fixed vs dynamic patching at 4x compression. This clearly shows that the performance improvements from dynamic patching are not tied to a specific backbone architecture and are transferable to any generic backbone.
>
> | Method                               | ETTh1 (MSE/MAE)      | ETTh2 (MSE/MAE)      | ETTm1 (MSE/MAE)      | ETTm2 (MSE/MAE)      | Weather (MSE/MAE)     |
> |--------------------------------------|-----------------------|-----------------------|-----------------------|-----------------------|------------------------|
> | Dynamic (avg patch size 4)  | 0.342 / 0.364       | 0.280 / 0.347       | 0.351 / 0.370       | 0.201 / 0.292       | 0.175 / 0.229        |
> | Fixed-size (size 4)    | 0.366 / 0.394       | 0.406 / 0.421       | 0.370 / 0.388       | 0.362 / 0.406       | 0.185 / 0.250        |

---

> ### Author Response · Authors · 2025-11-21
> **Response to reviewer foxx (Part 2)**
>
> **Questions:**
>
> **Q1: Analysis of patching mechanism**
> - We refer to Appendix Section G, where we visualize the learned patch boundaries on the evaluation datasets. The patterns align well with temporal dynamics: sudden level shifts or seasonality-driven changes produce **noticeable jumps** in adjacent observations, which are **immediately detected** by the relative-deviation criterion and result in new patch boundaries.
>
> - Within each patch, the SSM embeddings effectively capture the **intra-patch structure** (e.g., the regions between peaks and troughs). Due to the inductive bias of SSMs toward selective compression and long-range representation, they retain the relevant information **across these intermediate points**. The relative-deviation rule then identifies patch boundaries that minimize **information loss during compression**, enabling adaptive patching that is consistent across trends, seasonal components, and abrupt shifts.
>
> **Q2: Irregularly sampled series/inputs**
> - TimeSqueeze, as presented in this work, assumes **regularly sampled** inputs, consistent with the preprocessing pipelines used in recent time-series foundation models. However, the architecture itself does not fundamentally require evenly spaced data. Because dynamic patching relies on **relative deviation between consecutive observations**, it can operate on irregularly sampled series as long as the associated timestamps are provided.
>
> - Exploring a full irregular-time version of TimeSqueeze is an interesting extension, but the current paper focuses on the standard regularly sampled setting, aligned with existing TSFM benchmarks.
>
> **Q3: Suitability for financial time-series data**
> - Financial series often have long periods of low activity punctuated by sudden jumps or volatility spikes. Dynamic patching is naturally aligned with this structure: it allocates **small patches** in high-volatility regions to preserve sharp movements and **larger patches** in calm periods to avoid wasting tokens on redundant information. This allows the model to focus exactly where financial signals are most informative, without assuming stationarity or smoothness, making dynamic patching naturally suitable for financial data.

---

### Official Review · Reviewer_c3YR · 2025-11-01

**Soundness:** 2
**Presentation:** 4
**Contribution:** 2
**Rating:** 2
**Confidence:** 5

**Summary:**

The authors propose wrapping patch-based time series forecasting foundation models (focusing on Time-MoE) in a state space model encoder-decoder architecture that reduces the number of patches passed to the inner model. The experiments provided suggest that this does not significantly affect predictive metrics, but speeds up pretraining.

**Strengths:**

- The idea of wrapping a transformer model in an SSM encoder-decoder structure to speed it up is well-motivated and could have applications in other areas that use transformers.
- The paper is clear and well-written.
- Thorough ablations are provided.

**Weaknesses:**

- The time series forecasting models cited and evaluated against are out of date. Baseline evaluations are taken from Time-MoE (ICLR 2025) and have not been updated to include more recent models, such as Sundial [1] and Moirai-MoE [2], that have advanced the state-of-the-art in the meantime. As such, this paper's claim of state-of-the-art performance is not demonstrated. Posting models on the GIFT-Eval benchmark [3] has become common practice in this area, and many more recent performant models can be found there. (Ideally, the Chronos evaluations would also be updated to at least use Chronos-Bolt, released Nov. 2024.)
- The specific focus on Time-MoE in the method design and evaluations limits the impact of the paper, given the advances in the field since its publication. Without further discussion and evaluation, it's not necessarily clear that this method could be applied on top of more recent models and perform as well.
- Only five evaluation datasets are used - this is extremely limited and not in line with recent papers in this area. Even among earlier papers, MOMENT, TimesFM, Moirai, and Chronos all use dozens of datasets in their evaluations. Again, GIFT-Eval is an example of a large evaluation set that has become commonly used.
- A compelling explanation is lacking for placing patch boundaries based on the difference between neighbouring samples. For a seasonal input, this means focusing on the areas between peaks and troughs, but it's not clear why they should be treated as more relevant. Empirical justification could be all that's available, but it would strengthen the paper if some insight could be provided.
- It's not clear to me that it's very impactful to speed up pretraining of time series foundation models without improving other aspects of performance, given that they tend to be relatively cheap to train as far as foundation models go, and the zero-shot capability means pretraining only has to be done once. One possible benefit would be allowing more scaling of model size, but the results suggest that doing so does not help performance.

[1] Liu et al. "Sundial: A family of highly capable time series foundation models" ICML 2025.
[2] Liu et al. "Moirai-MoE: Empowering Time Series Foundation Models with Sparse Mixture of Experts" ICML 2025.
[3] https://huggingface.co/spaces/Salesforce/GIFT-Eval

**Questions:**

- Ablations compare to using a fixed patch length of 4 but the dynamic patching seems to prefer using patch length 2 in many cases - have you evaluated a fixed patch length of 2? (Acknowledging that this reduces the speedup benefits.)

---

> ### Author Response · Authors · 2025-11-21
> **Response to reviewer c3YR (Part 1)**
>
> We thank the reviewer for their feedback and address the concerns below.
>
> **Weaknesses:**
>
> **W1: Comparison with more baselines**
>
> - Thank you for highlighting an important concern regarding the baselines. We would like to clarify here that TimeSqueeze is indeed not “the state-of-the-art” in terms of forecasting, but is “one of the state-of-the-art”, when MSE/MAE performance is considered. With the rapid release of new time series foundation models (20+ new state-of-the-art models on GiftEval since July 2025), it is impractical to aim towards being the best in terms of MSE/MAE. Especially when most of the current models rely on proprietary as well as synthetic datasets, which are not available to the public, and also release only the inference code and not the training pipeline, making it impossible to reproduce the results and compare them in a fair manner. This applies to both [1] and [2], for which only the inference code is made available. Given that the **pretraining dataset, data selection, and the pretraining strategy** are equally important, if not more important, compared to the underlying architecture, direct comparison of MSE/MAE across models trained on different datasets is not entirely accurate.
>
> - Time-MoE is one of the few **open source** time series foundation models for which the full pretraining dataset and the pretraining pipeline are made publicly available. Hence, we integrate our idea of dynamic patching into the time-moe architecture, focusing on training the model from scratch to make a fair comparison and isolate the gains achieved by “dynamic patching”. Given that the Time-300B dataset may have a potential overlap with the GiftEval dataset, it is also not possible to evaluate TimeSqueeze or Time-MoE on GiftEval in a fair manner, and hence, we limit it to a long-term forecasting benchmark.
>
> **W2: Generalizability**
>
> Given that Time-MoE uses a generic decoder-only backbone + mixture of experts, and that dynamic patching operates purely at the input representation level, we expect the observed improvements to transfer to other modern architectures as well. Exploring these additional backbones is a valuable direction for future work.
>
> As additional evidence of generality, we replaced the entire Time-MoE-small backbone with a **generic 10M-parameter decoder-only Transformer** and conducted a controlled **ablation between fixed patching and dynamic patching**. This experiment removes any reliance on architectural details unique to Time-MoE and evaluates the patching mechanism on a completely different, lightweight backbone. As shown in the table below, dynamic patching still outperforms fixed patching even under this generic architecture, further supporting that the benefits of dynamic within-series patching arise from the method itself rather than from Time-MoE-specific design choices.
>
> | Method                               | ETTh1 (MSE/MAE)      | ETTh2 (MSE/MAE)      | ETTm1 (MSE/MAE)      | ETTm2 (MSE/MAE)      | Weather (MSE/MAE)     |
> |--------------------------------------|-----------------------|-----------------------|-----------------------|-----------------------|------------------------|
> | Dynamic patching (avg patch size 4)  | 0.342 / 0.364       | 0.280 / 0.347       | 0.351 / 0.370       | 0.201 / 0.292       | 0.175 / 0.229        |
> | Fixed-size patching (size 4)    | 0.366 / 0.394       | 0.406 / 0.421       | 0.370 / 0.388       | 0.362 / 0.406       | 0.185 / 0.250        |
>
> **W3: Limited eval datasets**
>
> - In the TSFM setting, benchmarking is heavily influenced by differences in training corpus, preprocessing, and pipeline design, which makes direct comparisons between independently trained foundation models less informative about our specific contribution. For this reason, our primary focus is on the **relative improvements introduced by dynamic patching**, keeping the backbone architecture fixed as Time-MoE so that any performance gain can be attributed solely to the patching mechanism, and not claiming the absolute state-of-the-art forecasting performance.
>
> - Time-MoE is one of the few open source time series foundation models for which the full pretraining dataset and the pretraining pipeline is made publicly available and hence our choice of backbone for implementing the dynamic patching mechanism. But because of the potential **overlap between Time-300B dataset and GiftEval datasets**, it is not possible to fairly evaluate TimeSqueeze on Time-300B dataset and instead choose the LTSF benchmark, which was the choice for the original TimeMoE paper.

---

> ### Author Response · Authors · 2025-11-21
> **Response to reviewer c3YR (Part 2)**
>
> **Weaknesses (contd.):**
>
> **W4: Dynamic patching vs. Seasonal inputs**
>
> - The relative-deviation criterion is designed to detect points of significant local change, where a new patch should begin so that the SSM embedding at the boundary can faithfully **summarize the upcoming segment**. In regions with rapid variations, peaks, troughs, or sharp transitions, the deviation between neighboring samples is large relative to the local power. A single SSM embedding cannot adequately capture these fluctuations, so the method places boundaries more frequently, allocating more representational capacity to information-rich parts of the signal.
>
> - Conversely, in smooth regions where the signal changes slowly, the deviation is small relative to the local power. Consecutive samples contribute little new information, and a single SSM embedding can **summarize a long stretch of the sequence**. This naturally leads to larger patch sizes in redundant areas.
>
> - For seasonal inputs, this behavior is intuitive: the criterion identifies not only the major peaks and troughs but also intermediate rises or dips within each period, precisely the regions where local dynamics matter most. The **flatter intervals** between these transitions are **easier to compress without losing fidelity**, making the overall patching strategy effective and aligned with the intrinsic structure of seasonal time series. Thus, the relative-deviation metric is useful for identifying **compression-critical points**, locations where summarizing many steps into a single embedding would lose significant local information. Importantly, this does not imply that these points are inherently “more relevant” for the forecasting task. The SSM encoder continues to preserve information from all intermediate samples, peak or non-peak, through its **inductive bias** and recurrent structure. The deviation metric simply guides where compression can safely occur without degrading the fidelity of the SSM representations, rather than signaling semantic forecasting-importance in the underlying time series.
>
>
> **W5: Importance of improving pretraining efficiency**
>
> - We would like to highlight that the pretraining time and memory cost remain major practical barriers for time-series foundation models, especially outside of large industry labs. As evidenced by the GiftEval benchmark, only a handful of open-source teams trained competitive models, and the largest Time-MoE variant required **128 A100-80GB GPUs**, far beyond what is realistically accessible to most academic groups.
>
> - In our own experiments, we were unable to train the TimeSqueeze-large model for a sufficient number of epochs due to **GPU limitations**, which is why its gains over TimeSqueeze-small are not as pronounced as expected. Dynamic patching directly addresses this bottleneck by reducing both **training time and memory footprint**, allowing substantially larger models to be trained on modest hardware.
>
> - Thus, even if pretraining is done once, lowering its computational and memory requirements is highly **impactful**: it increases accessibility, enables broader experimentation, and brings large-scale TSFM research closer to the academic community.
>
> **Questions:**
>
> **Q1: Additonal ablation w.r.t fixed-patch size 2**
>
> - We trained an additional baseline using a fixed patch size of 2. Its performance is slightly worse than the fixed patch size of 4, as shown in the table below. We believe this is due to two factors: (i) very small fixed patches limit the model’s ability to capture meaningful local structure, and (ii) the resulting increase in the number of tokens substantially raises the sequential modeling burden for the transformer backbone. In contrast, dynamic patching selects small patches only when needed, for instance in high-variation regions, and uses larger patches elsewhere, achieving a more favorable **balance** between resolution and efficiency.
>
>
> | Model / Variation                 | ETTh1 (MSE/MAE)   | ETTh2 (MSE/MAE)   | ETTm1 (MSE/MAE)   | ETTm2 (MSE/MAE)   | Weather (MSE/MAE)   |
> |----------------------------------|--------------------|--------------------|--------------------|--------------------|----------------------|
> | TimeSqueeze_base                | 0.357 / 0.384      | 0.281 / 0.336      | 0.311 / 0.343      | 0.181 / 0.270      | 0.166 / 0.216        |
> | TimeSqueeze w/ fixed patch 4     | 0.373 / 0.396      | 0.455 / 0.448      | 0.359 / 0.382      | 0.335 / 0.380      | 0.178 / 0.232        |
> | TimeSqueeze w/ fixed patch 2     | 0.376 / 0.402   | 0.475 / 0.462    | 0.361/ 0.384    | 0.357 / 0.396    | 0.179 / 0.233     |

---

> > ### Comment · Reviewer_c3YR · 2025-11-24
> >
> > Thank you for the detailed response. My concerns with the paper still generally have to do with the limited set of evaluations failing to make it clear that this method is an important contribution. If the purpose of the paper is not to compete with state-of-the-art models (which I think would require changes to the writing) and instead to show improvements when other aspects of the architecture and pretraining are held constant, I think evidence is needed that this method generalizes across current architectures and pretraining approaches, with respect to a range of evaluation tasks. Otherwise, the practical impact of the work could be very limited. Currently, the paper demonstrates that improvements are seen on five long-term forecasting tasks for a specific architecture that is no longer state-of-the-art. While I appreciate the additional results you've shared, I still feel that there's a wide gap between them and results that would indicate an impactful contribution.
> >
> > **W1**
> > While I appreciate the challenges of keeping up with new releases in this field, from what I can tell, all baselines are taken from an ICLR 2025 paper with no newer baselines run, making it a year out of date. I don't see that as being adequate for making judgments about how this model compares to the state of the art. The statement in the abstract, "Extensive experiments demonstrate that TimeSqueeze achieves state-of-the-art forecasting performance", also seems to make a stronger claim than the position you've provided here.
> >
> > GIFT-Eval comparisons could be done by training your model on a non-overlapping dataset such as the GiftEvalPretrain dataset. A faster although less comprehensive comparison would be to evaluate your model on the datasets without overlaps, and compare to other models on those datasets, since the GIFT-eval repo contains results broken down by model and dataset.
> >
> > **W2**
> > These results do provide some more confidence about the method, although generalizing across more current competitive architectures/pretraining methods is really what would show that this method is useful.
> >
> > **W3**
> > I don't think this addresses the gap between this paper and the other papers in the field I mentioned. Aside from having a wider range of evaluation datasets, evaluating on short-horizon forecasting is also a common way to extend the breadth of experiments.
> >
> > **W4**
> > I could see an argument for distributing information more evenly across patches, but information/compressibility and differences between consecutive samples are two separate quantities and it's easy to come up with cases where they're unrelated. For instance, a sine wave with additive white noise has the same entropy at every point, but will have regions of larger and smaller differences.
> >
> > **W5**
> > While research accessibility is valuable, this is still a field with much lower computational requirements than other foundation models, and the computational costs have not prevented the rapid development of many new state-of-the-art models (as you noted above). But for practitioners, this is typically not a significant issue, since foundation models can be used with no further training. (The Time-MoE example is somewhat of an outlier - more recent models have not generally followed its approach of extremely high parameter counts.)

---

### Official Review · Reviewer_YLk2 · 2025-11-01

**Soundness:** 2
**Presentation:** 3
**Contribution:** 2
**Rating:** 4
**Confidence:** 4

**Summary:**

This paper proposes a hybrid forecasting architecture that combines the strengths of point-wise and patch-based embeddings through dynamic time series compression. It comprises a lightweight state-space encoder that uses point-wise embeddings to process full-resolution time series and extract fine-grained temporal features. An adaptive patching module then prunes these features using variable-sized patches, assigning smaller patches to information-rich regions and larger patches to redundant segments.

**Strengths:**

S1. This paper presents a hybrid forecasting architecture to incorporate dynamic, content-aware patching for adaptive compression in time series.

  S2. The experimental findings validate the computational efficiency of the proposed method.

**Weaknesses:**

1. Time series data often exhibit periodic and trend patterns. Relying solely on single-step differences between adjacent samples to determine boundaries may be insufficient for capturing periodic boundaries or trend changes.


   2. The patching mechanism determines boundaries by comparing the absolute difference between adjacent samples with the local average power within a sliding window. Could the authors clarify how this criterion effectively distinguishes between information-rich and redundant regions?


   3. The boundary selection depends on the design of the sliding window, yet the paper does not clearly specify whether the window is overlapping or non-overlapping.


   4. Experimental results show that model performance decreases as the compression ratio increases, which may be due to excessive information loss caused by over-compression. Intuitively, using a smaller compression ratio might improve performance, but the paper does not provide corresponding experiments.



   5. The current experimental results show limited forecasting performance, and the paper does not include comparisons with recent Time Series Foundation Models (TSFMs), such as Sundial, LightGTS, and VisionTS.



   6. The paper lacks an analysis of patch distribution across different datasets. It would be valuable to examine how patch length, density, or boundary frequency vary among datasets with different statistical characteristics.

**Questions:**

See Weaknesses.

---

> ### Author Response · Authors · 2025-11-21
> **Response to reviewer YLk2 (Part 1)**
>
> We thank the reviewer for their feedback and address the concerns in detail below.
>
> **Weaknesses:**
>
> **W1: Justification for 1-step relative deviation metric for patching**
>
> - We would like to clarify that the purpose of our boundary selection is **not** to detect periodic or trend changes directly. Patch boundaries are computed to optimize the **information retained in the SSM embeddings**, not to identify semantic structural shifts in the raw signal.
>
> - In Stage 1, the input sequence is processed at full resolution by the SSM encoder, whose inductive bias naturally captures **trends, periodicity, and long-range patterns** across the entire sequence. Thus, all Stage-1 embeddings already encode these temporal structures.
>
> - In Stage 2, the relative-deviation criterion is used only to identify which embeddings can be **safely compressed or merged without significant loss of information**, effectively performing a lossy token compression guided by the SSM’s learned representation. Therefore, the dynamic patching mechanism operates on top of the rich temporal structure already captured by the encoder, rather than relying on single-step differences to detect periodic or trend boundaries.
>
> **W2: Relative deviation vs Information rich regions**
>
> - The relative-deviation criterion is specifically designed to locate **points of significant local change**, where a new patch should begin so that the SSM embedding at the patch boundary can adequately summarize the upcoming region.
>
> - In information-rich regions, where the signal exhibits rapid or large changes, the deviation between adjacent samples is high relative to the local power. A single SSM embedding **cannot fully represent these rich variations**, so patch boundaries are placed more frequently.
>
> - Conversely, when the signal is locally smooth, the deviation is small relative to the local power. In these redundant regions, consecutive samples provide little new information, and a **single SSM embedding** is sufficient to summarize many time steps. This naturally leads to larger patch sizes. Thus, the relative-deviation rule effectively adapts patch sizes based on whether new information is being introduced, distinguishing rich regions from redundant ones.
>
> **W3: Overlapping vs non-overlapping sliding window**
>
> - The sliding window considered for boundary detection is **overlapping**, as described by the equation in section 3.1.2. We will update the section to explicitly mention this to avoid any confusion.
>
> **W4: Comparison with lower compression rate**
>
> - We now include results for a smaller compression ratio **($2 \times$)**. The updated table shows:
>
> | Compression Rate | $2 \times$     | $4 \times$     | $6 \times$     | $8 \times$     |
> |------------------|--------|--------|--------|--------|
> | Average MSE      | 0.272  | 0.261  | 0.297  | 0.313  |
>
> - These results indicate that $2 \times$ compression is slightly worse than $4 \times$, while $4 \times$ remains better than both $6 \times$ and $8 \times$. Taken together, this suggests that the model achieves its best **balance between compression efficiency and information preservation around $4 \times$** compression. This empirical pattern supports the idea that there is a **optimal compression rate**, beyond which over-compression becomes harmful and under-compression offers little gain.

---

> ### Author Response · Authors · 2025-11-21
> **Response to reviewer YLk2 (Part 2)**
>
> **W5: Concerns about limited forecasting performance**
>
> - In the TSFM setting, benchmarking is heavily influenced by differences in **training corpus, preprocessing, and pipeline design**, which makes direct comparisons between independently trained foundation models **less informative** about our specific contribution. For this reason, our primary focus is on the **relative improvements introduced by dynamic patching**, keeping the backbone architecture fixed as Time-MoE so that any performance gain can be attributed solely to the patching mechanism.
>
> - But as per the suggestion, we now provide zero-shot evaluations of TimeSqueeze alongside Sundial, VisionTS, and LightGTS to provide a more complete picture. As shown in the updated table, TimeSqueeze performs comparably to these baselines on some datasets but worse on others, which is potentially because of the fact that these models differ in **both scale and training pipelines**.  Specifically for recent time-series foundation models (MOIRAI, Time-MOE, Sundail) their performance improvements correlates with the increase in their pretraining  dataset size: LOTSA (MOIRAI): 27 Billion,  Time-300B: 300 Billion, Sundail: **1 Trillion points**. None of these works compare their architecture performance on a **different dataset**. It is possible that dataset is a crucial factor of their improvements in performance.
>
> | Method       | ETTh1 (MSE/MAE) | ETTh2 (MSE/MAE) | ETTm1 (MSE/MAE) | ETTm2 (MSE/MAE) | Weather (MSE/MAE) |
> |--------------|------------------|------------------|------------------|------------------|---------------------|
> | Sundial      | 0.390 / 0.418    | 0.340 / 0.387    | 0.354 / 0.388    | 0.265 / 0.324    | 0.233 / 0.271       |
> | VisionTS     | 0.390 / 0.414    | 0.333 / 0.375    | 0.374 / 0.372    | 0.282 / 0.321    | 0.269 / 0.292       |
> | LightGTS     | 0.401 / 0.424    | 0.362 / 0.397    | 0.345 / 0.378    | 0.249 / 0.318    | 0.219 / 0.266       |
> | TimeSqueeze  | 0.402 / 0.416    | 0.363 / 0.408    | 0.417 / 0.412    | 0.292 / 0.349    | 0.257 / 0.293       |
>
> **W6: Patch-size distribution**
>
> - We included an analysis of patch-size distributions across all evaluation datasets in Appendix Section G and H. As shown, patch statistics naturally reflect the underlying signal characteristics: datasets with smoother temporal dynamics, such as Weather, produce **larger average patch sizes**, while datasets with more rapidly varying patterns, such as ETTm2, result in significantly **smaller patches**. This confirms that the dynamic patching mechanism adapts meaningfully to the local variability of each dataset.

---

### Official Review · Reviewer_Yq4K · 2025-11-01

**Soundness:** 2
**Presentation:** 2
**Contribution:** 2
**Rating:** 4
**Confidence:** 4

**Summary:**

This paper proposes TimeSqueeze, a dynamic patching architecture for efficient long-context time series forecasting. The model addresses the trade-off between fine-grained temporal resolution and computational scalability. TimeSqueeze introduces a two-stage hybrid representation: 1. A lightweight state-space encoder extracts fine-grained temporal features from the full-resolution time series. 2. An adaptive patching module dynamically adjusts patch sizes, assigning smaller patches to regions with complex temporal variations and larger ones to stable segments. This variable-resolution representation allows the Transformer backbone to process fewer tokens without losing critical information, improving both efficiency and accuracy.

**Strengths:**

1. The methodology is well-motivated and clearly integrated into the forecasting framework.

2. The paper is clearly written and conceptually intuitive.

**Weaknesses:**

1. The experimental validation is limited, as the evaluations are conducted only on the Time-MoE architecture, which restricts the generality of the conclusions.

2. The overall architecture of TimeSqueeze largely builds upon the Time-MoE framework — equations (2–4) are directly inherited from the original Time-MoE paper — and the idea of dynamic patching has already been explored in several prior works.

3. The efficiency comparison with Time-MoE is not entirely fair, since Time-MoE is intentionally designed to maximize model capacity by using point-wise rather than patch-based embeddings. Therefore, a more appropriate efficiency analysis should include lightweight baselines such as SparseTSF, TimeBase, or DLinear.

4. The full-shot forecasting experiments lack strong state-of-the-art baselines such as CycleNet, TQNet, TimeBase, or DUET, which makes it difficult to assess the claimed superiority of the proposed model.

**Questions:**

Have you considered extending the TimeSqueeze architecture to handle multi-dimensional time series data, such as those involving spatial-temporal correlations?

---

> ### Author Response · Authors · 2025-11-21
> **Response to reviewer Yq4K (Part 1)**
>
> We thank the reviewer for their feedback and address the concerns below.
>
>  **Weaknesses:**
>
> **W1: Generalizability of conclusions**
> - Our primary goal in this work is to isolate the **benefit of dynamic patching** compared to point embeddings and fixed-length patching under **controlled, identical training conditions**. To do this, we build on Time-MoE, one of the few state-of-the-art time-series foundation models with fully open-sourced datasets, preprocessing, and training pipelines. This allows us to integrate our SSM-based dynamic patching directly into the same backbone and train all variants from scratch under the exact same data, compute, and optimization settings, ensuring a fair comparison in both forecasting accuracy and pretraining efficiency.
>
> - Given that Time-MoE uses a decoder-only backbone + mixture of experts, and that dynamic patching operates purely at the input representation level, we expect the observed improvements to transfer to other modern architectures as well. Exploring these additional backbones is a valuable direction for future work.
>
> - As additional evidence of generality, we replaced the entire Time-MoE-small backbone with a **generic 10M-parameter decoder-only Transformer (no mixture of experts)** and conducted a controlled ablation between fixed patching and dynamic patching. This experiment removes any reliance on architectural details unique to Time-MoE and evaluates the patching mechanism on a **completely different, lightweight backbone**. As shown in the table below, dynamic patching still outperforms fixed patching even under this generic architecture, further supporting that the benefits of dynamic within-series patching arise from the method itself rather than from Time-MoE-specific design choices.
>
> | Method                               | ETTh1 (MSE/MAE)      | ETTh2 (MSE/MAE)      | ETTm1 (MSE/MAE)      | ETTm2 (MSE/MAE)      | Weather (MSE/MAE)     |
> |--------------------------------------|-----------------------|-----------------------|-----------------------|-----------------------|------------------------|
> | Dynamic patching (avg patch size 4)  | 0.362 / 0.383       | 0.280 / 0.347       | 0.351 / 0.370       | 0.201 / 0.292       | 0.175 / 0.229        |
> | Fixed-size patching (size 4)    | 0.366 / 0.394       | 0.406 / 0.421       | 0.370 / 0.388       | 0.362 / 0.406       | 0.185 / 0.250        |
>
> **W2: Novelty of dynamic patching**
> - We acknowledge that our backbone follows the Time-MoE formulation; this choice is intentional, as it provides a strong and transparent foundation on which to isolate the effect of our contribution. However, while prior work has explored series-level dynamic patching (i.e., choosing different fixed patch sizes per input), these methods **do not adapt patch boundaries within a sequence**.  Please see the discussion with **reviewer GKgg** where we distinguish TimeSqueeze from the methods HDMixer, IMTS, and LightGTS.
>
> - TimeSqueeze is, to our knowledge, the first to introduce **state-space–driven dynamic patching that varies patch size along the time axis within an input**, allowing the model to allocate finer resolution to complex regions and coarser resolution to smooth regions. This “within-series” adaptive patching is fundamentally different from previous approaches and is the core novelty of our method.
>
> **W3: Efficiency comparisons**
> - We focused our efficiency analysis on the **foundation-model and zero-shot forecasting** setting, where architectures like SparseTSF, TimeBase, and DLinear are not directly comparable in terms of efficiency, as they are not designed as large pretrain–finetune models. Our goal is **not** to claim the best absolute efficiency, but to measure the relative gains that dynamic patching brings over point embeddings under a **fixed foundation-model backbone**. Time-MoE provides the appropriate setting for this comparison, as it is a strong, open-source TS foundation model with a matching training pipeline. We expect the same relative efficiency and accuracy benefits of dynamic patching to transfer to lighter backbones as well, which we see as a promising direction for future work.

---

> ### Author Response · Authors · 2025-11-21
> **Response to reviewer Yq4K (Part 2)**
>
> **Weaknesses (contd.):**
>
> **W4: Addional comparisons for full-shot forecasting**
> - We have now expanded our comparison to include TimeBase, CycleNet, TQNet, and DUET, all trained fully on the train splits of each evaluation dataset. For fairness, TimeSqueeze and Time-MoE are pretrained on the large pretraining corpus and then finetuned for one epoch on the same train splits. We present the avg. MSE/MAE for prediction horizons {96, 192, 336, 720} for all models below.
>
> | Method          | ETTh1 (MSE/MAE) | ETTh2 (MSE/MAE) | ETTm1 (MSE/MAE) | ETTm2 (MSE/MAE) | Weather (MSE/MAE) |
> |-----------------|------------------|------------------|------------------|------------------|---------------------|
> | TimeBase        | 0.396 / 0.415    | 0.347 / 0.398    | 0.357 / 0.381    | 0.251 / 0.314    | 0.219 / 0.263       |
> | CycleNet/Linear | 0.432 / 0.427    | 0.383 / 0.404    | 0.386 / 0.395    | 0.272 / 0.315    | 0.254 / 0.279       |
> | CycleNet/MLP    | 0.457 / 0.441    | 0.388 / 0.409    | 0.379 / 0.396    | 0.266 / 0.314    | 0.243 / 0.271       |
> | TQNet           | 0.441 / 0.434    | 0.378 / 0.402    | 0.377 / 0.393    | 0.277 / 0.323    | 0.242 / 0.269       |
> | DUET            | 0.398 / 0.419    | 0.334 / 0.383    | 0.338 / 0.369    | 0.248 / 0.307    | 0.218 / 0.252       |
> | TimeSqueeze     | 0.398 / 0.419    | 0.350 / 0.393    | 0.383 / 0.386    | 0.259 / 0.321    | 0.243 / 0.279       |
>
> - As shown in the updated table, TimeSqueeze outperforms TQNet and both CycleNet variants, and performs comparably to or slightly below TimeBase and DUET. This positioning is consistent with expectations: TimeSqueeze is designed as a foundation-model approach with strong zero-shot and few-shot capabilities, whereas TimeBase and DUET are **specialized full-shot models optimized directly on train split of eval datasets**. Nevertheless, TimeSqueeze achieves competitive full-shot accuracy while also offering substantial advantages in long-context modeling and zero-shot generalization for the time series foundation model.
>
> - We would also like to clarify that achieving the absolute best full-shot forecasting accuracy is not the primary goal of this work. Full-shot performance is heavily influenced by the specific pretraining corpus, dataset composition, and training dynamics, which vary widely across TSFM pipelines. Instead, our focus is on introducing the **first fully dynamic patching framework for time-series foundation models** and demonstrating that it provides consistent efficiency and accuracy gains over equivalent point and fixed-patch embedding approaches under controlled conditions. We believe this contribution is broadly applicable and can improve the computational and practical efficiency of any TSFM backbone, independent of the underlying architecture or dataset choice.
>
> **Questions**
>
> **Q1: Extension to spatio-temporal correlations**
> - In this work, we follow the **channel-independence** strategy of PatchTST, to transform a multivariate input into univariate series, allowing TimeSqueeze to handle **any-variate** series.  This setup already allows us to handle high-dimensional multivariate inputs, but it does not explicitly encode structured spatial correlations (e.g., grid or graph structure).
>
> - That said, the proposed dynamic patching mechanism is **orthogonal to the choice of spatial encoder** and can be extended to multi-dimensional spatio-temporal data. A natural extension is to apply a spatial module, such as CNN, at each time step to capture spatial dependencies, and then apply TimeSqueeze along the temporal axis of the resulting latent sequences, or to design joint spatio-temporal patches over space-time blocks. We view such spatio-temporal extensions as an exciting direction for future work.

---

> ### Comment · Reviewer_Yq4K · 2025-11-25
>
> Thank you for the detailed rebuttal. However, I still have some concerns. First, the level of innovation remains limited, as the method builds incrementally upon Time-MoE and the idea of dynamic  patching has been explored in works such as PatchMLP and others [2]. Second, the claimed efficiency advantages are not yet convincingly demonstrated. It seems that it is not more efficiently than previous efficient forecasting methods. Third, the forecasting performance still lags behind strong baselines like DUET on several datasets, suggesting notable room for improvement. Given these remaining concerns, I will maintain my current score at this stage, while continuing to focus on the discussion between the authors and other reviewers for further evaluation.
>
> [1] Enhancing Time Series Forecasting through Selective Representation Spaces: A Patch Perspective

---

### Official Review · Reviewer_D1Uq · 2025-11-02

**Soundness:** 3
**Presentation:** 3
**Contribution:** 3
**Rating:** 6
**Confidence:** 3

**Summary:**

TimeSqueeze presents a well-executed and impactful approach to efficient long-context time-series modeling via dynamic patching. Its hybrid design and strong empirical results make it a valuable contribution. However, the work could be strengthened by broader comparisons with adaptive compression methods and a more in-depth analysis of the learned representations.

**Strengths:**

1. The introduction of TimeSqueeze, which dynamically combines point-level fine-grained encoding with adaptive patch-level compression, is a novel and well-motivated. The dynamic patching mechanism based on relative deviation effectively addresses the limitations of fixed-size patching and enables content-aware compression.

2. The paper demonstrates compelling efficiency gains (up to 20× faster training and 10× faster inference) while maintaining competitive forecasting performance with state-of-the-art point-embedding models like Time-MoE in both zero-shot and full-shot settings across multiple benchmarks.

3. The authors provide extensive experiments, including comparisons with strong baselines, detailed ablation studies, and analyses of the impact of pre-trained context length and compression rates, which convincingly validate the design choices and scalability of the proposed method.

**Weaknesses:**

1. While the paper compares with fixed-patching methods and point-embedding models, it does not include comparisons with other adaptive or learned compression strategies from recent literature (e.g., learned chunking or entropy-based methods), leaving the relative advantage of the proposed patching criterion less fully contextualized.
2. Although the paper shows that longer pre-trained contexts improve performance, the analysis is limited to performance curves without deeper investigation into what temporal structures or dependencies are better captured, or how the dynamic patching interacts with long-range modeling.

**Questions:**

Please see weaknesses!

---

> ### Author Response · Authors · 2025-11-21
> **Response to reviewer D1Uq**
>
> We thank the reviewer for their appreciation of our work and address the weaknesses in detail below:
>
> **Weaknesses:**
>
> **W1: Comparison with other compression strategies**
>
> - Thank you for pointing out the connection to learned chunking and entropy-based adaptive compression. These techniques have been developed almost entirely in the context of **language models**, where data live in a discrete token space and token-wise entropy is well defined. Applying such methods to **continuous-time series data** would require additional discretization layers and would require that entropy models be trained separately, making end-to-end optimization and efficient runtime challenging.
>
> - In contrast, current time-series foundation models, including TimeSqueeze, operate directly in continuous signal space, where we can leverage natural **notions of local distance and variance**. This is exactly what our deviation-based criterion exploits, **without relying on tokenization or a secondary scoring model**. Learned chunking in NLP also depends heavily on similarity in learned token embeddings, which can be brittle and highly sensitive to the tokenizer; time-domain signals have a much more stable and meaningful numeric structure.
>
> - To the best of our knowledge, TimeSqueeze is the **first approach to perform dynamic patching within a signal for time-series foundation models**, adapting patch size based on local signal statistics rather than discrete tokens or external importance models.
>
> - Finally, we agree that ideas from learned chunking and entropy-based compression are promising for time-series foundation models more broadly, as already mentioned this in the Conclusion and plan to explore these directions in future work.
>
>
> **W2: Analysis for long-context pretraining**
>
> - We have clearly shown that exposure to longer sequences during pretraining translates into better downstream performance under a fixed inference context. The results in Figure 3(b) show that this effect is consistent across datasets: models pretrained with longer contexts learn temporal representations that transfer well even when the inference context length is capped at 512.
>
> - For a deeper analysis, we now break down the performance at a dataset level and observations suggest that the gains are most pronounced on datasets with strong seasonal or long-range patterns, which indicates that the model is better able to internalize slower dynamics when trained on extended histories. Dynamic patching supports this by preserving fine-grained detail in locally complex regions while still allowing the model to see much longer sequences overall.
>
> Following is comparison for inference context length 512 and prediction horizon 96
>
> | Pretraining Context | ETTh1 (MSE/MAE) | ETTh2 (MSE/MAE) | ETTm1 (MSE/MAE) | ETTm2 (MSE/MAE) | Weather (MSE/MAE) |
> |---------------------|------------------|------------------|------------------|------------------|--------------------|
> | **2048**            | 0.359 / 0.385    | 0.282 / 0.346    | 0.312 / 0.344    | 0.181 / 0.275    | 0.167 / 0.217      |
> | **1024**            | 0.398 / 0.427    | 0.313 / 0.384    | 0.343 / 0.378    | 0.190 / 0.289    | 0.180 / 0.234      |

---

> > ### Comment · Reviewer_D1Uq · 2025-11-25
> > **Response to authors**
> >
> > Dear Authors,
> >
> > Thank you for the author's further clarification. I have decided to keep my positive score.

---

### Meta-Review · Area_Chair_BLqJ · 2025-12-20

**Summary:**

TimeSqueeze addresses an efficiency bottleneck in long-context time series forecasting. The reviewers provided highly mixed feedback on this paper. Two reviewers expressed positive evaluations, recognizing the practical value of TimeSqueeze in reducing computational overhead for transformer-based time series forecasting while largely preserving accuracy. In particular, they appreciated the hybrid design that combines point-wise representations via a lightweight state-space encoder with adaptive, content-aware patching to compress long contexts efficiently. However, the majority of reviewers raised substantial concerns regarding novelty, experimental coverage, and empirical rigor. A central recurring issue is that the proposed dynamic patching mechanism is perceived as incremental relative to a growing body of prior work on adaptive or content-aware patching, and that the paper does not sufficiently articulate or empirically demonstrate how TimeSqueeze differs fundamentally in mechanism or benefit from these approaches. Several reviewers also questioned whether the empirical gains are convincingly demonstrated, given the limited number of datasets, the heavy focus on a single backbone (Time-MoE), and the absence of comparisons against more recent time series foundation models and lightweight forecasting baselines.

Additional concerns span multiple dimensions, including insufficient analysis of patching behavior, sensitivity to hyperparameters, lack of frequency- or pattern-level interpretability, unclear generalization to multivariate or spatio-temporal settings, and limited discussion of theoretical trade-offs between compression and information loss. While rebuttal responses clarified some design motivations and provided additional explanations, for most reviewers these clarifications did not fully resolve the underlying concerns.

Overall, while the paper presents a practically motivated and potentially useful efficiency-oriented framework, reviewers remain on whether the contribution rises to the level of a clear and broadly impactful advance over existing methods. Thus, I believe this research cannot be accepted at this time.

**Reviewer Concerns:**

The authors provided extensive rebuttal experiments and discussions, and address partial concerns, may include 1) The efficiency motivation of dynamic patching for long-context forecasting is generally accepted, and multiple reviewers agree that reducing transformer input length without severe performance degradation is a relevant and worthwhile goal. 2) The hybrid design combining point-wise SSM encoding with patch-based transformer processing is seen by some reviewers as a reasonable engineering choice that balances fidelity and efficiency. 3) Clarifications in the rebuttal helped explain the authors’ intended scope, namely improving efficiency–performance trade-offs while holding other architectural components fixed, rather than claiming universal state-of-the-art forecasting accuracy.

However, based on evaluation, these four concerns may still remain,

**1) Limited novelty and differentiation from prior work.** Multiple reviewers (Yq4K, c3YR, GKgg) argue that dynamic or content-aware patching has already been explored in several recent works, and that the paper does not convincingly demonstrate a fundamentally new mechanism or insight beyond these approaches. Despite rebuttal clarifications, this concern remains central.

**2) Experimental scope and baseline selection.** The evaluation is widely viewed as insufficiently comprehensive. Key issues include reliance on a single backbone (Time-MoE), omission of strong and recent baselines (e.g., DUET, Sundial, Moirai-MoE, LightGTS), and the use of only five datasets, which is far below current community standards for TSFM evaluation.

**3) Fairness of efficiency comparisons.** Reviewers question comparisons against point-embedding models that were not designed for efficiency, and note the absence of comparisons with lightweight or patch-based baselines, weakening claims of superior efficiency.

**4) Understanding and justification of the patching criterion.** Several reviewers raise doubts about using local point-wise differences as a proxy for information density, particularly for periodic, trending, or highly nonstationary series. Empirical or theoretical justification remains limited.

**Reviewer Scores:**

TimeSqueeze addresses an efficiency bottleneck in long-context time series forecasting and is viewed positively by two reviewers as a practical hybrid design. But the majority of reviewers remain unconvinced that the work demonstrates sufficient novelty, experimental breadth, or generality to justify acceptance at this venue. The unresolved concerns regarding baseline coverage, evaluation scale, and differentiation from prior dynamic patching methods prevent a clear consensus in favor of acceptance.

---

### Decision · Program_Chairs · 2026-01-26

Reject